# LDP: A Lightweight Denoising Plugin Enhancing Generalization in Single-Image Super-Resolution

## Abstract

Current single-image super-resolution (SISR) models struggle to generalize to real-world degradations. To address this challenge, we propose LDP, an innovative lightweight denoising autoencoder (DAE) plug-in. It improves the generalization ability of SR models via low-resolution (LR) images prediction-based cyclic regularization. LDP models the SISR degradation process within the DAE framework. It leverages a property of diffusion models, where after noise is added, high-resolution (HR) images and LR features become aligned, so that denoising noisy HR features is equivalent to denoising noisy LR features. During the corruption process, noise is added independently to each HR patch. During the denoising process, a convolutional denoiser uses learned filters to approximate blur kernels. In addition, LR degradation is used to distinguish different LR from the same HR. LDP can be applied to SR models in two modes: as a training loss to improve reconstruction quality, or as an inference post-processing step to correct artifacts. Extensive experiments demonstrate that LDP substantially improves the generalization of existing SR models to unseen degradations.

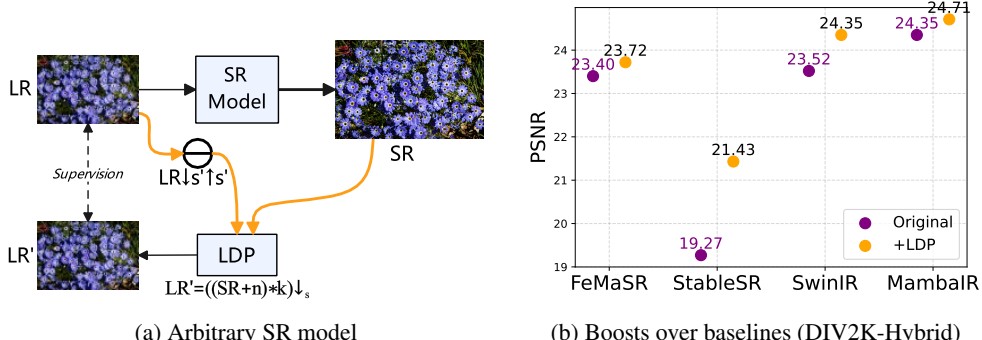

(a) Arbitrary SR model      (b) Boosts over baselines (DIV2K-Hybrid)

Figure 1: Our LDP is a lightweight denoising autoencoder-based plug-in that can be seamlessly integrated into arbitrary SR models, operating as a training-time loss or an inference-time module.

## 1 Introduction

Single Image Super-Resolution (SISR) aims to reconstruct high-resolution (HR) images from their low-resolution (LR) counterparts. SISR is widely applied in various fields, such as medical imaging Li et al. (2024a) and remote sensing Dong et al. (2024). Deep learning has advanced SISR architectures from Convolutional Neural Network (CNN) Dong et al. (2014) to Transformer Liang et al. (2021); Chen et al. (2023b) and State-Space Model Guo et al. (2024; 2025), achieving higher reconstruction accuracy. Meanwhile, generative methods, including Generative Adversarial Network (GAN) Chen et al. (2022) and Diffusion Model Wang et al. (2024); Yue et al. (2025); Zhang et al. (2025), have been explored to improve perceptual quality.

Despite advances in SR architectures, existing models struggle to generalize to unseen degradations. Recent approaches leverage data augmentation and self-supervised learning techniques to tackle this challenge. Data augmentation approaches typically fall into two categories: generating synthetic distortions Zhang et al. (2021a); Wang et al. (2021), or employing generative models Li et al. (2022); Chen et al. (2025) to synthesize paired data from unpaired LR and HR images. However, these methods may harm performance Zhang et al. (2023) or are limited to in-distribution datasets. Self-supervised approaches rely on either image-specific training Shocher et al. (2018); Ulyanov et al. (2018) or test-time adaptation Hussein et al. (2020); Zhou et al. (2023); Chen et al. (2024), utilizing internal image statistics and priors. However, they suffer from high computational cost or the need for model-specific adaptation. Addressing unseen degradations efficiently remains a key challenge.

To address these limitations, we propose LDP, a lightweight denoising autoencoder (DAE) plug-in. It improves the generalization ability of SR models via LR prediction-based cyclic regularization. LDP models the SISR degradation process within the DAE framework. It leverages a property of diffusion models, where after noise is added, high-resolution (HR) images and LR features become aligned Wang et al. (2023b), making denoising noisy HR features equivalent to denoising noisy LR features. LDP takes high-resolution images (ground-truth HR or SR outputs) as input for degradation modeling, with LR high-frequency components as a condition to distinguish different LR images from the same HR. During the corruption process, LDP introduces patch-dependent Gaussian noise. This enables the model to learn fine-grained degradation in local patches, rather than assuming the same degradation for the whole image. During the denoising process, a lightweight convolutional denoiser learns the blur kernels associated with the degradation model. Built on these designs, LDP accurately generates corresponding LR image and generalizes well to unseen degradations. LDP applies to SR models in two modes: as a training-time loss function to improve reconstruction quality, or as an inference-time post-processing step that corrects artifacts independently of training. Extensive experiments verify that LDP significantly improves the generalization ability of existing SR models on unknown complex degradations.

Overall, our contributions are three-fold:

- We propose LDP, an innovative lightweight denoising autoencoder plug-in for single-image super-resolution that enhances the generalization of existing SR models.

- LDP is a conditional degradation model that generates LR images from HR inputs by explicitly conditioning on LR high-frequency components. LDP operates in two modes: as a degradation-aware training-time loss function, or as an inference-time correction module (e.g., Posterior Sampling for diffusion models).

- LDP enhances reconstruction quality during training as a loss function and mitigates artifacts at inference independently of training. Both modes improve SR model generalization to unknown complex degradations.

## 2 RELATED WORK

### 2.1 IMPROVING GENERALIZATION IN SR

The limited generalization ability of SR models to unseen degradations remains a major challenge for real-world applications. Existing SR methods address this issue using two main approaches: data augmentation and self-supervised learning. Data augmentation methods seek to bridge the training–inference gap by creating synthetic data with degradations that approximate real-world scenarios. One line of works explicitly model degradations using predefined operations. BSR-GAN Zhang et al. (2021a) generates complex degradations by sequentially combining downsampling, blur, noise, and compression in random order, producing varied LR images for training. RealESRGAN Wang et al. (2021) introduces higher-order degradations to reflect real-world degradation chains. While BSRGAN and RealESRGAN enable non-blind SR models to handle blind scenarios through multi-degradation training, such strategies may compromise performance on in-distribution benchmarks Zhang et al. (2023). Alternatively, implicit modeling methods leverage generative models to synthesize paired data from real LR and unpaired HR images. GAN Yuan et al. (2018); Li et al. (2022); Yin et al. (2023) or diffusion-based Chen et al. (2025) methods learn degradation priors to create realistic training pairs. However, their generalization remains limited to in-distribution data. Self-supervised learning enables SISR training using only LR images without

paired HR supervision. ZSSR Shocher et al. (2018) and DIP Ulyanov et al. (2018) exploit internal patterns or implicit priors without external data. CorrectFilter Hussein et al. (2020); Zhou et al. (2023) aligns inputs with the training distribution of pre-trained models. Lway Chen et al. (2024) uses a degradation model to synthesize LR images from SR outputs for test-time fine-tuning. Although effective, these methods are computationally expensive or require model-specific adaptation.

## 2.2 Constraining the SR Solution Space via Degradation Modeling

Degradation modeling, applied jointly with the SR model, introduces structural constraints that ensure reconstructed LR outputs align with the LR input, effectively narrowing the solution space to favor LR-consistent reconstructions. DRN Guo et al. (2020) adds a degradation branch that projects SR outputs back to the LR domain, enforcing reconstruction consistency and improving stability. DualSR Emad et al. (2021) introduces a dual-path framework where a GAN-based downsampler and an upsampler are jointly trained with cycle consistency to model and reverse image-specific degradations. SCL-SASR Chen et al. (2023a) adopts a similar bidirectional design under MAP estimation, coupling SR and degradation networks to adapt to test-time degradations. Lway Chen et al. (2024) introduces test-time adaptation with pre-trained degradation models to fine-tune SR models, increasing generalization to unseen degradations. Despite their benefits, these methods face several limitations: DRN handles only bicubic downsampling; DualSR and SCL-SASR require image-specific optimization or joint training; and Lway introduces significant computational overhead due to its large model size. In contrast, our method supports a wide range of degradations through an explicitly modeled degradation process within a lightweight denoising autoencoder framework. Our degradation modeling framework is adaptable to various training settings, from large-scale supervised learning to image-specific fine-tuning, and can also be applied directly at test time. The framework is lightweight and does not incur significant computational cost.

Degradation modeling is also applied during inference in diffusion-based image restoration to enforce LR consistency. ILVR Choi et al. (2021) guides the sampling process of DDPM Ho et al. (2020) using a reference image to maintain low-frequency consistency across the denoising steps. DR2 Wang et al. (2023b) shows that injecting additional Gaussian noise makes LR and HR distributions less distinguishable, allowing noise-corrupted LR images to be treated as noise-corrupted HR images during sampling. MCG Chung et al. (2022) ensures samples stay close to the data manifold by projecting the gradient of the measurement function onto its tangent space. DPS Chung et al. (2023) further leverages the degradation process to connect the LR observation to the predicted clean image at each step. In our method, LDP degrades each predicted clean image during diffusion inference, treating it as SR to produce a predicted LR image. We then enforce LR cyclic consistency by applying the tailored loss $\mathcal{L}_{\text{sym}}^{\text{FT}}$ (Eq. 16), which penalizes the discrepancy between the predicted LR and the ground-truth LR. This degradation-aware constraint enhances fidelity by suppressing artifacts and promoting structural consistency in the SR results.

## 3 Proposed Method

Section 3.1 outlines the motivation behind LDP. Section 3.2 introduces the overall framework of LDP. Section 3.3 then details its training and inference modes, describing LDP's own training, its application in fine-tuning SR models, and its role as a post-processing step for diffusion models.

### 3.1 Motivation

To improve the generalization of existing SR models on unknown complex degradations, we adopt a degradation modeling approach applied jointly with the SR model. This introduces structural constraints that ensure the reconstructed LR outputs are aligned with the LR input, effectively narrowing the solution space to favor LR-consistent reconstructions. Our LDP integrates degradation modeling Yue et al. (2022) into the denoising autoencoder, reinterpreting denoising as a controllable degradation applied to HR images. In the classical degradation formulation, this can be expressed as:

$$y = ((x + n) \otimes k) \downarrow_s, \tag{1}$$

where $x \in \mathbb{R}^{H \times W \times 3}$ is the HR image, $y \in \mathbb{R}^{\frac{H}{s} \times \frac{W}{s} \times 3}$ is the LR image, $n$ is the noise, $k$ is the blur kernel, and $s$ is the downsampling scale. We further leverage a property of diffusion models,

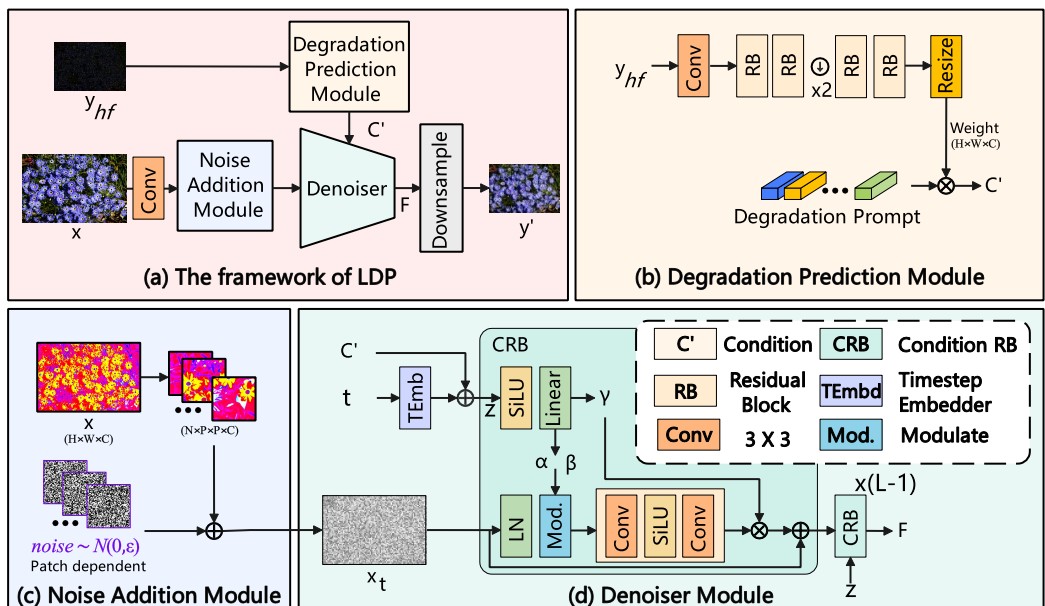

Figure 2: **(a) LDP Framework.** $LR_{hf}$ predicts degradation $C'$, guiding the noise-perturbed HR features to generate the LR output via denoising and downsampling. **(b) Degradation Prediction.** Stacked RB generate weights from $LR_{hf}$ and multiply them with $P_D$ to produce $C'$. **(c) Noise Addition.** Patch-dependent noise is added to HR features at random timesteps. **(d) Denoiser.** A lightweight CNN denoises $HR_t$ conditioned on $z$ using CRBs with AdaLN.

whereby after noise is added, HR features and LR features become aligned Wang et al. (2023b), making denoising noisy HR features equivalent to denoising noisy LR features. This allows us to perform degradation modeling on HR images using a denoising autoencoder. However, there remains a challenge: since the SR task is inherently ill-posed, a condition is required to differentiate between different LR images generated from the same HR image under varying degradations. This condition must satisfy three criteria: (1) it cannot be the LR image itself, otherwise the network might take shortcuts and fail to learn meaningful degradations; (2) it must be discriminative for different LR images corresponding to the same HR image; and (3) it should be simple and easy to obtain. We define this condition as $LR_{hf}$, obtained by subtracting the $s'$-fold downsampled-then-upsampled LR image from the original LR image. In summary, we use a denoising autoencoder to perform degradation modeling on the input HR image, with the condition $LR_{hf}$ controlling the type of degradation in the output. During application, this approach constrains the super-resolution (SR) model to produce outputs whose LR reconstructions (via our LDP) are consistent with the original LR input, thus enforcing LR cyclic consistency and effectively guiding the SR model.

## 3.2 FRAMEWORK

Figure 2 (a) illustrates the framework of our proposed LDP, which consists of four main modules: the Degradation Prediction Module (DPM), Noise Addition Module (NAM), Denoiser Module and Downsample Module. Designed as a denoising autoencoder, LDP functions as a conditional degradation model that generates LR images from HR inputs by conditioning on LR high-frequency components. To facilitate both implementation and interpretability, we adopt the noise corruption process from diffusion models Ho et al. (2020). The overall process of LDP is formulated as:

$$x_t = NAM(x, t), \tag{2}$$

$$y' = D(Denoiser(x_t|DPM(y_{hf}), t)), \tag{3}$$

Where $y'$ is the predicted LR images, and $y_{hf}$ is the LR high-frequency component. $t$ is a patch-dependent timestep, $x_t$ is the noised HR features, $NAM(\cdot)$ is the Noise Addition Module, $DPM(\cdot)$ is the Degradation Prediction Module and $D(\cdot)$ is the Downsample Module.

**Degradation Prediction Module.** Figure 2 (b) shows the DPM diagram. Its input is the high-frequency component of the LR image, computed by subtracting the s'-fold downsampled-then-upsampled LR image from the original LR image, which can be formulated as:

$$y_{hf} = y - y \downarrow_{s'} \uparrow_{s'}, \tag{4}$$

where $\downarrow_{s'}$ and $\uparrow_{s'}$ denote the downsampling and upsampling operations with scale factor $s'$, respectively. To extract degradation information, we use prompts to encode degradation-specific details Potlapalli et al. (2023). First, a weight map $w$ is derived from $y_{hf}$, and then resized to match the spatial dimensions of $x$ (i.e., $H \times W$). This resized weight map is multiplied element-wise with the Degradation Prompt $P_D$. It forms a degradation map $C' \in \mathbb{R}^{H \times W \times C}$ and serves as the condition for the denoiser. The process can be formulated as:

$$w = (\text{RB}_4 \circ \text{RB}_3 \circ \downarrow_2 \circ \text{RB}_2 \circ \text{RB}_1) \circ Conv(y_{hf}), \tag{5}$$

$$C' = P_D \otimes \text{Resize}(w, H, W), \tag{6}$$

where $\text{RB}(\cdot)$ denotes a residual block consisting of two $3 \times 3$ convolutional layers with a SiLU activation in between, $Conv(\cdot)$ represents a convolutional layer, $\circ$ denotes function composition applied sequentially from right to left, and $\otimes$ denotes element-wise multiplication. The downsampling operator $\downarrow_2$ further reduces spatial resolution and disrupts local structures. The degradation prompt $P_D \in \mathbb{R}^{N_p \times C}$ is jointly learned to encode degradation-specific information.

**Noise Addition and Denoiser Module.** Our framework integrates degradation modeling Yue et al. (2022) into the denoising autoencoder, reinterpreting denoising as a controllable degradation applied to HR images. During the corruption process, we perturb HR images using a patch-wise noise schedule. Specifically, following the diffusion noise schedule, each patch $x_i \in \mathbb{R}^{P \times P \times C}$ is assigned a random timestep $t_i$, and its noisy version is obtained as:

$$x_i^{(t_i)} = \sqrt{\hat{\alpha}_{t_i}} \, x_i + \sqrt{1 - \hat{\alpha}_{t_i}} \, \epsilon_i, \quad \epsilon_i \sim \mathcal{N}(0, \mathbf{I}), \tag{7}$$

where $\hat{\alpha}_{t_i}$ denotes the cumulative product of noise scheduling coefficients at time $t_i$ and $\epsilon_i$ is standard Gaussian noise. This patch-wise formulation enables each image region to undergo a different level of degradation, allowing the model to better capture spatially varying corruption. The final noisy image is denoted as $x_t$.

During the denoising process, a lightweight CNN acting as the denoiser module estimates the blur kernel and extracts intermediate feature $F$ conditioned on the degradation map $C'$. The feature $F$ are then downsampled to produce the predicted LR image. Specifically, the denoiser module comprises $L$ Condition Residual Blocks (CRBs) that leverage Adaptive Layer Normalization (AdaLN) Perez et al. (2018); Li et al. (2024b) for conditional modulation. For each $P \times P$ patch, the assigned timestep $t_i$ is embedded and combined with $C'$ to produce a patch-specific condition $z$. This condition is passed through a SiLU activation and a linear layer to generate modulation parameters $\alpha$, $\beta$, and $\gamma$ corresponding to scaling, bias, and gating. In the residual path, features are first normalized via LayerNorm and modulated by $\alpha$ and $\beta$, then processed by a residual block, gated with $\gamma$, and finally added back to the input. The CRB can be formulated as:

$$t_{emb} = TEmb(t), \tag{8}$$

$$\alpha, \beta, \gamma = Linear(SiLU(C' + t_{emb})), \tag{9}$$

$$x'_t = \alpha \otimes (LN(F_{i-1})) + \beta, \tag{10}$$

$$F_i = \gamma \otimes RB(x'_t) + F_{i-1}, \tag{11}$$

where $TEmb(\cdot)$ is the timestep embedder, $F_{i-1}$ is the output of the previous CRB, and the initial feature is set as $F_0 = x_t$. The $RB(\cdot)$ in the final CRB is simplified to a single convolutional layer.

**Downsample Module.** The module adjusts the feature map to match the spatial resolution of the original LR image. Features $F$ are first downsampled by a factor of $s$, then processed by a residual block and a convolutional layer:

$$y' = Conv(\text{RB}(F \downarrow_s)). \tag{12}$$

Here, $\text{RB}$ and the final convolutional layer are used to enhance feature representation and maintain smooth transitions between downsampled regions.

### 3.3 Training and Inference Modes of LDP

**Training LDP.** Following Lway Chen et al. (2024), LDP is trained by supervising only the high-frequency components of the predicted LR images. We apply the Discrete Wavelet Transform (DWT) to decompose the predicted LR image $y'$ into four subbands (LL, LH, HL, HH). The high-frequency subbands (LH, HL, HH) are then summed and normalized to form a weight map $M$, which is subsequently used to compute both the L1 loss and the LPIPS loss Zhang et al. (2018):

$$\mathcal{L}_{sym}^{T} = \lambda_1 \mathcal{L}_1(M \otimes y', M \otimes y) + \lambda_2 \mathcal{L}_{LPIPS}(M \otimes y', M \otimes y), \tag{13}$$

where $\lambda_1$ and $\lambda_2$ are the corresponding loss weights.

**Fine-Tuning SR Models with LDP.** In fine-tuning, the original loss of pretrained SR models is augmented with a frequency loss Xie et al. (2023) that supervises the amplitude and phase components of SR and HR images in the frequency domain:

$$\mathcal{L}_{fre} = \frac{1}{HW} \sum_{u=0}^{H-1} \sum_{v=0}^{W-1} D(\mathcal{F}(x')(u,v), \mathcal{F}(x)(u,v)), \tag{14}$$

$$D(\mathcal{F}(x'), \mathcal{F}(x)) = \left( (\mathcal{R}(\mathcal{F}(x')) - \mathcal{R}(\mathcal{F}(x)))^2 + (\mathcal{I}(\mathcal{F}(x')) - \mathcal{I}(\mathcal{F}(x)))^2 \right)^{\gamma/2}, \tag{15}$$

where $x$ and $x'$ are the HR image and SR result, $\mathcal{F}(x)$ denotes the 2D Fourier transform of $x$, and $\mathcal{R}(\cdot)$ and $\mathcal{I}(\cdot)$ denote its real and imaginary parts. $\gamma$ controls the sharpness of the frequency distance and is set to 1 by default. $(u, v)$ indexes the frequency domain. In addition, LDP enforces cycle consistency by reconstructing the LR image from the SR output and minimizing a symmetric loss:

$$\mathcal{L}_{sym}^{FT} = \lambda_1 \mathcal{L}_1(M' \otimes y', M' \otimes y) + \lambda_2 \mathcal{L}_{LPIPS}(M' \otimes y', M' \otimes y) + \lambda_3 \mathcal{L}_{fre}(M' \otimes y', M' \otimes y), \tag{16}$$

where $M' = \tau \cdot M$, $\tau$ scales the high-frequency weight map $M$ by a scalar $\tau$.

**Diffusion Posterior Sampling with LDP.** Our LDP can also be applied during inference in diffusion models via Diffusion Posterior Sampling (DPS) Chung et al. (2023), which uses the gradient of a data fidelity term to guide sampling and better align the results with the LR input:

$$\nabla_{\boldsymbol{x}_t} \log p_t(x_t|y) \simeq \boldsymbol{s}_{\theta*}(x_t, t) - \rho \nabla_{x_t} \mathcal{L}_{sym}^{FT}(LDP(\hat{x}_0, y_{hf}), y), \tag{17}$$

where $\boldsymbol{s}_{\theta*}(x_t, t)$ denotes the score function (the noise predictor in DDPM Ho et al. (2020)), and $LDP(\cdot)$ represents our LDP degradation model. $\hat{x}_0$ denotes the predicted clean image at each time step, and we treat it as the SR output. In latent diffusion models, $\hat{x}_0$ is first decoded into the pixel space before computing the gradient.

## 4 Experiment

### 4.1 Implementation Details

**Training LDP.** We train LDP on LSDIR Li et al. (2023) dataset using BSRGAN Zhang et al. (2021a) to synthesize diverse degradation datasets. For a scale factor of $s = 4$, the key hyperparameters are $s' = 2$, $L = 3$, $P = 16$, $N_p = 32$, $\lambda_1 = \lambda_2 = 1$, and $C = 64$, resulting in 642k parameters. We use the Adam Kingma & Ba (2015) optimizer with $\beta_1 = 0.9$ and $\beta_2 = 0.99$, with a fixed learning rate of 0.001. The batch size is 12, with $256 \times 256$ HR patches. The timesteps $t_i$ are sampled from $[500, 1000]$ to align the noisy HR and LR features. We adopt the diffusion batch multiplier Li et al. (2024b) with a value of 4 to perform multiple noise realizations per HR image. Training is conducted on a single NVIDIA RTX A6000 for 60K iterations, taking approximately 16 hours.

**Fine-Tuning SR Models.** We fine-tune existing SR models on the DF2K dataset (DIV2K Agustsson & Timofte (2017) and Flickr2K Lim et al. (2017)) using BSRGAN degradation patterns, with our LDP employed as an auxiliary loss. Details are provided in the Appendix D.

**Testing.** For synthetic testing, we generate five distinct datasets from the DIV2K validation set using bsrgan_plus (BSRGAN Zhang et al. (2021a) and Real-ESRGAN Wang et al. (2021)), corresponding to the following degradation types: (1) downsampling, (2) noise, (3) blur, (4) JPEG compression, and (5) hybrid degradations following bsrgan_plus defaults. For real-world testing,

Table 1: Performance of multiple degradation models in LR prediction on synthetic multi-degradation datasets.

| Methods | Metrics | Down | Noise | Blur | JPEG | Hybrid |
|---|---|---|---|---|---|---|
| **DRN** | PSNR↑ | **32.05** | **27.25** | 26.38 | **29.65** | 27.03 |
| | SSIM↑ | **0.9539** | 0.7812 | 0.8273 | **0.9270** | 0.8098 |
| | LPIPS↓ | **0.0794** | 0.2474 | 0.3207 | **0.0826** | 0.3360 |
| **DualSR** | PSNR↑ | 19.58 | 18.77 | 19.36 | 18.57 | 19.36 |
| | SSIM↑ | 0.4814 | 0.4712 | 0.4911 | 0.4612 | 0.4883 |
| | LPIPS↓ | 0.1408 | 0.1399 | 0.1844 | 0.1492 | 0.2130 |
| **LDP** | PSNR↑ | 29.15 | 26.71 | **28.41** | 28.01 | **27.94** |
| | SSIM↑ | 0.9283 | **0.8978** | **0.9159** | 0.9243 | **0.9173** |
| | LPIPS↓ | 0.0985 | **0.1248** | **0.1417** | 0.0877 | **0.1025** |

Table 2: Similarity between the LR images generated by multiple degradation models and the downsampled SR images.

| Methods | Metrics | Down | Noise | Blur | JPEG | Hybrid |
|---|---|---|---|---|---|---|
| **DRN** | PSNR↑ | 34.02 | 31.57 | 34.99 | 31.35 | 35.10 |
| | SSIM↑ | 0.9638 | 0.9590 | 0.9692 | 0.9587 | 0.9679 |
| | LPIPS↓ | 0.0365 | 0.0436 | 0.0306 | 0.0467 | 0.0296 |
| **DualSR** | PSNR↑ | 22.58 | 20.79 | 22.57 | 20.46 | 22.85 |
| | SSIM↑ | 0.6689 | 0.6502 | 0.7044 | 0.6356 | 0.7164 |
| | LPIPS↓ | 0.1264 | 0.1040 | 0.1262 | 0.1279 | 0.1175 |
| **LDP** | PSNR↑ | 28.41 | 25.93 | 25.04 | 27.42 | 26.28 |
| | SSIM↑ | 0.8895 | 0.7508 | 0.7596 | 0.8886 | 0.7597 |
| | LPIPS↓ | 0.1551 | 0.3043 | 0.3278 | 0.1293 | 0.3586 |

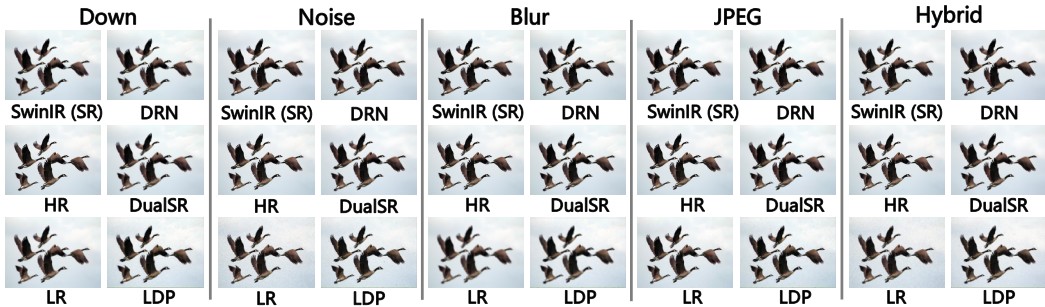

Figure 3: Qualitative results of multiple degradation models for LR prediction on synthetic datasets. (**Zoom in for details**)

we evaluate on RealSR Cai et al. (2019), RealSRSet Zhang et al. (2021b), and DPED Ignatov et al. (2017) datasets. We evaluate using PSNR, SSIM Wang et al. (2004), and LPIPS Zhang et al. (2018) as reference metrics, and NIQE Mittal et al. (2012), MANIQA Yang et al. (2022), CLIPIQA Wang et al. (2023a), MUSIQ Ke et al. (2021), and QAlign Wu et al. (2024) as non-reference metrics. For diffusion models, synthetic datasets are center-cropped to $512 \times 512$, and real-world datasets follow the StableSR Wang et al. (2024).

## 4.2 EFFECTIVENESS OF LDP IN LR PREDICTION

To thoroughly evaluate the effectiveness of the proposed LDP, we conduct extensive experiments under five degradation scenarios and compare it with two existing degradation models, DRN Guo et al. (2020) and DualSR Emad et al. (2021). In this experiment, we first generate SR images using SwinIR Liang et al. (2021), and then apply the degradation models provided by LDP, DRN, and DualSR to obtain predicted LR images from the SR outputs. These predictions are compared with the LR inputs to the SR model, and the results are reported in Table 1. In addition, Table 2 reports the similarity between the LR images produced by each degradation model and the downsampled SR images. A higher similarity indicates that the degradation model collapses into trivial downsampling rather than applying the specific degradations implied by the input LR. As shown in the tables, LDP performs consistently well across all degradation types. Importantly, the similarity between the LDP-generated LR and the downsampled SR is significantly lower than that between the LDP-generated LR and the input LR, demonstrating that LDP does not degenerate into simple downsampling. In contrast, DRN behaves almost identically to bicubic downsampling: because its inputs include only HR (SR results) images without any conditional signals, it fails to map an SR image to the multiple possible LR variants implied by different degradations. DualSR also struggles to properly handle diverse degradation types, particularly under complex mixed settings. As illustrated in **Fig.** 3, LDP effectively degrades high-frequency structures, further validating its ability to generate perceptually realistic LR images even under challenging degradations. In contrast, DRN and DualSR largely produce LR outputs that resemble simple downsampled versions of the SR images, indicating that they fail to apply the intended degradations.

Table 3: Performance improvements of blind SR models across diverse architectures using our proposed LDP on synthetic multi-degradation benchmarks. We generate synthetic benchmarks from the DIV2K validation set using five types of degradation: (1) Downsampling (Down), (2) Noise, (3) Blur, (4) JPEG, and (5) Hybrid degradations following bsrgan_plus defaults.

| Datasets | Scale | Metrics | FeMaSR | +LDP | StableSR | +LDP | SwinIR | +LDP | MambaIR | +LDP |
|---|---|---|---|---|---|---|---|---|---|---|
| **Down** | ×4 | PSNR↑ | 24.22 | **25.06** (+0.84) | 20.35 | **21.73** (+1.38) | 25.44 | **25.86** (+0.42) | 26.58 | **26.63** (+0.05) |
| | ×4 | SSIM↑ | 0.6793 | **0.7105** (+0.0312) | 0.4998 | **0.5642** (+0.0644) | 0.7210 | **0.7242** (+0.0032) | 0.7393 | **0.7403** (+0.0010) |
| | ×4 | LPIPS↓ | 0.2637 | **0.2490** (-0.0147) | 0.3746 | **0.2870** (-0.0876) | 0.2579 | **0.2538** (-0.0041) | 0.2054 | **0.2005** (-0.0049) |
| **Noise** | ×4 | PSNR↑ | 22.82 | **23.84** (+1.02) | 19.95 | **21.48** (+1.53) | 24.34 | **25.04** (+0.70) | 26.11 | **26.34** (+0.23) |
| | ×4 | SSIM↑ | 0.6519 | **0.6957** (+0.0438) | 0.4569 | **0.5599** (+0.1030) | 0.7130 | **0.7198** (+0.0068) | 0.7382 | **0.7411** (+0.0029) |
| | ×4 | LPIPS↓ | 0.2788 | **0.2624** (-0.0164) | 0.4279 | **0.3040** (-0.1239) | 0.2676 | **0.2659** (-0.0017) | 0.2279 | **0.2219** (-0.0060) |
| **Blur** | ×4 | PSNR↑ | 24.12 | **24.42** (+0.30) | 19.98 | **21.50** (+1.52) | 24.03 | **24.67** (+0.64) | 24.99 | **25.33** (+0.34) |
| | ×4 | SSIM↑ | 0.6639 | **0.6787** (+0.0148) | 0.4373 | **0.5437** (+0.1064) | 0.6764 | **0.6833** (+0.0069) | 0.6892 | **0.6942** (+0.0050) |
| | ×4 | LPIPS↓ | **0.3168** | 0.3199 (+0.0031) | 0.5112 | **0.4763** (-0.0349) | 0.3197 | **0.3168** (-0.0029) | 0.2768 | **0.2751** (-0.0017) |
| **JPEG** | ×4 | PSNR↑ | 22.92 | **23.87** (+0.95) | 20.17 | **21.91** (+1.74) | 24.55 | **25.27** (+0.72) | 26.36 | **26.59** (+0.23) |
| | ×4 | SSIM↑ | 0.6696 | **0.7068** (+0.0372) | 0.5141 | **0.5943** (+0.0802) | 0.7301 | **0.7372** (+0.0071) | 0.7497 | **0.7538** (+0.0041) |
| | ×4 | LPIPS↓ | 0.2633 | **0.2508** (-0.0125) | 0.3682 | **0.2767** (-0.0915) | 0.2535 | **0.2506** (-0.0029) | 0.2113 | **0.2063** (-0.0050) |
| **Hybrid** | ×4 | PSNR↑ | 23.40 | **23.72** (+0.32) | 19.27 | **21.43** (+2.16) | 23.52 | **24.35** (+0.83) | 24.35 | **24.71** (+0.36) |
| | ×4 | SSIM↑ | 0.6211 | **0.6392** (+0.0181) | 0.3656 | **0.5197** (+0.1541) | 0.6458 | **0.6492** (+0.0034) | 0.6587 | **0.6636** (+0.0049) |
| | ×4 | LPIPS↓ | **0.3453** | 0.3516 (+0.0063) | 0.5727 | **0.4461** (-0.1266) | 0.3634 | **0.3571** (-0.0063) | 0.3244 | **0.3210** (-0.0034) |

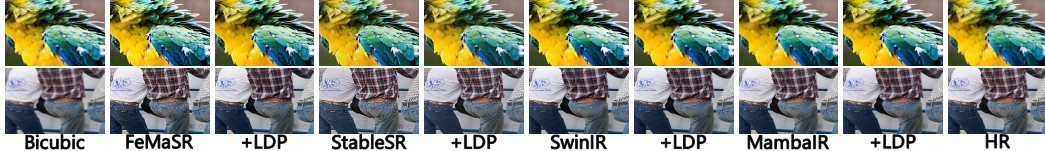

|  Bicubic | FeMaSR | +LDP | StableSR | +LDP | SwinIR | +LDP | MambaIR | +LDP | HR |

Figure 4: Qualitative results on synthetic datasets with ×4 scale factor. (**Zoom in for details**)

### 4.3 IMPROVING EXISTING SR MODELS VIA FINE-TUNING WITH LDP

We evaluate LDP on Blind SR models, including the GAN-based FeMaSR Chen et al. (2022), Diffusion-based StableSR Wang et al. (2024), Transformer-based SwinIR Liang et al. (2021), and Mamba-based MambaIR Guo et al. (2024). In these experiments, LDP is applied only during the fine-tuning stage and is not used at inference.

**Improving SR Models on Synthetic Benchmarks.** Quantitative and qualitative results are presented in Tab. 3 and Fig. 4 (Fig. 7 in **Appendix**). As listed in Tab. 3, incorporating LDP consistently improves all baseline models across all degradation types. Among them, MambaIR+LDP achieves the best overall performance. SwinIR and StableSR also benefit significantly from LDP. StableSR, in particular, shows substantial relative gains under challenging conditions such as Blur and Hybrid. These results highlight LDP's effectiveness in narrowing the solution space via cycle consistency, enabling stronger generalization to unknown degradations. Although FeMaSR+LDP outperforms the original model in most metrics, its LPIPS values in Blur and Hybrid remain higher. As shown in Fig. 4, LDP effectively reduces GAN artifacts and corrects texture distortions, significantly improving perceptual quality. The low LPIPS scores of the original FeMaSR are likely due to severe GAN artifacts misinterpreted as texture.

**Improving SR Models on Real-World Benchmarks.** Quantitative and qualitative results are presented in Tab. 4 and Fig. 5 (Fig. 8 in **Appendix**). Table 4 shows that incorporating LDP consistently improves the performance of existing blind SR models across almost all datasets and metrics, demonstrating its enhanced generalization to unseen degradations. For FeMaSR, LDP suppresses GAN-induced artifacts, producing more stable, natural outputs. This can lower no-reference metrics, e.g., the CLIPIQA score drops on RealSR, as such metrics may favor visually striking but structurally inaccurate results. As shown in Fig. 5, the visual results explain the numerical improvements, with LDP mitigating ringing and GAN-induced artifacts, thereby enhancing visual fidelity and contributing to the better no-reference metrics scores.

### 4.4 LDP FOR POSTERIOR SAMPLING OF PRETRAINED DIFFUSION MODELS

We evaluated how LDP enhances pre-trained diffusion models through posterior sampling, including LDM Rombach et al. (2022), StableSR Wang et al. (2024), ResShift Yue et al. (2025), and UPSR Zhang et al. (2025). Quantitative and qualitative results are presented in Tab. 5 and

Table 4: Performance improvements of blind SR models across diverse architectures using our proposed LDP on real-world benchmarks.

| Datasets | Scale | Metrics | FeMaSR | +LDP | StableSR | +LDP | SwinIR | +LDP | MambaIR | +LDP |
|---|---|---|---|---|---|---|---|---|---|---|
| RealSR | ×4 | NIQE↓ | **4.708** | 5.533 (+0.825) | 7.446 | **6.331** (-1.115) | **4.773** | 4.838 (+0.065) | **5.330** | 5.350 (+0.020) |
| | ×4 | MANIQA↑ | 0.3430 | **0.3654** (+0.0224) | 0.3303 | **0.3548** (+0.0245) | 0.3510 | **0.3742** (+0.0232) | 0.2882 | **0.3374** (+0.0492) |
| | ×4 | CLIPIQA↑ | **0.5645** | 0.4482 (-0.1163) | 0.4886 | **0.5213** (+0.0327) | 0.4739 | **0.5478** (+0.0739) | 0.3989 | **0.4642** (+0.0653) |
| | ×4 | MUSIQ↑ | 58.94 | **60.70** (+1.76) | 52.99 | **59.26** (+6.27) | 59.67 | **61.91** (+2.24) | 51.87 | **57.85** (+5.98) |
| | ×4 | QAlign↑ | 3.695 | **3.860** (+0.165) | 2.347 | **2.646** (+0.299) | 3.820 | **3.877** (+0.057) | 3.631 | **3.766** (+0.135) |
| DPED | ×4 | NIQE↓ | **5.045** | 5.704 (+0.659) | 7.616 | **7.228** (-0.388) | 4.982 | **4.821** (-0.161) | 5.983 | **5.430** (-0.553) |
| | ×4 | MANIQA↑ | **0.3102** | 0.2719 (-0.0383) | **0.3056** | 0.2970 (-0.0086) | 0.2637 | **0.2832** (+0.0195) | 0.2334 | **0.2767** (+0.0433) |
| | ×4 | CLIPIQA↑ | **0.5570** | 0.3610 (-0.1960) | **0.3968** | 0.3843 (-0.0125) | 0.3402 | **0.4538** (+0.1136) | 0.3083 | **0.3850** (+0.0767) |
| | ×4 | MUSIQ↑ | 49.14 | 44.07 (-5.07) | 42.97 | **45.08** (+2.11) | 42.10 | **45.91** (+3.81) | 35.25 | **44.64** (+9.39) |
| | ×4 | QAlign↑ | **3.429** | 3.262 (-0.167) | 2.033 | **2.311** (+0.278) | 2.988 | **3.090** (+0.102) | 3.192 | **3.248** (+0.056) |
| RealSRSet | ×4 | NIQE↓ | **5.236** | 5.952 (+0.716) | 6.090 | **5.586** (-0.504) | **5.424** | 5.441 (+0.017) | **5.726** | 5.893 (+0.167) |
| | ×4 | MANIQA↑ | **0.4006** | 0.4002 (-0.0004) | 0.3904 | **0.4012** (+0.0108) | 0.3740 | **0.3938** (+0.0198) | 0.2978 | **0.3555** (+0.0577) |
| | ×4 | CLIPIQA↑ | **0.6874** | 0.5683 (-0.1191) | 0.6057 | **0.6214** (+0.0157) | 0.5843 | **0.6376** (+0.0533) | 0.4793 | **0.5428** (+0.0635) |
| | ×4 | MUSIQ↑ | 64.65 | 64.07 (-0.58) | 60.15 | **62.84** (+2.69) | 63.60 | **65.33** (+1.73) | 55.96 | **61.28** (+5.32) |
| | ×4 | QAlign↑ | 3.776 | **3.870** (+0.094) | 2.916 | **3.247** (+0.331) | 2.749 | **3.322** (+0.573) | 3.434 | **3.632** (+0.198) |

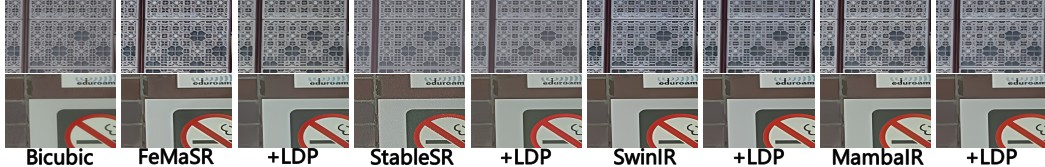

Bicubic  FeMaSR  +LDP  StableSR  +LDP  SwinIR  +LDP  MambaIR  +LDP

Figure 5: Qualitative results on real-world benchmarks with ×4 scale factor. (**Zoom in for details**)

Fig. 6 (Fig. 9 in **Appendix**). As listed in Tab. 5, after applying LDP, the baselines show improvements across nearly all metrics on most datasets. For instance, StableSR demonstrates notable gains in MANIQA, CLIPIQA, and MUSIQ scores after applying LDP, while ResShift and UPSR also achieve higher metric values in most cases. For StableSR, we applied the noise-subtraction technique (Appendix E), which accounts for the differences from Tab. 4. As showed in Fig. 6, our LDP effectively reduces texture artifacts while preserving structural consistency.

## 5 ABLATION STUDY

In ablation study, we examine the loss components, patch size, frequency band selection, scale factor for high-frequency acquisition, performance of LDP under severe degradations, and computational burden of LDP. Further details are provided in Appendix F.

**Ablation of Loss Terms in the Fine-Tuning Stage.** Table 6 presents the impact of different loss components in $\mathcal{L}_{sym}^{FT}$ (Equ. 16) and $\mathcal{L}_{fre}$ (Equ. 14) during fine-tuning of pretrained SwinIR models, evaluated on the synthetic Hybrid dataset. In all experiments, we set $\tau = 100$ and the weight of each loss term is set to 1. All variants using any combination of the proposed losses outperform the baseline. Incorporating both symmetric and frequency losses (LDPV5–LDPV7) consistently improves perceptual quality (lower LPIPS) and reconstruction accuracy (higher PSNR and SSIM), with LDPV7 achieving the best overall performance, highlighting the complementary nature of these loss components. The LDP parameters can be universally configured as $\tau = 100$ and $\lambda_1 = \lambda_2 = \lambda_3 = 1$ for any super-resolution model, leading to improved generalization performance.

**Ablation of the weight of** $tau$**.** Table 7 presents the impact of different weight of $tau$ when fine-tuning SwinIR. All values of $tau$ outperform the baseline, with $tau = 100$ achieving the best overall performance.

## 6 LIMITATIONS AND CONCLUSION

We propose LDP, a lightweight denoising autoencoder plug-in. By integrating HR images and the high-frequency component of LR, the model achieves realistic degradation modeling while maintaining efficiency. Experiments show LDP significantly improves the generalization of existing SR models on unseen degradations after fine-tuning, and enables test-time artifact correction. However, LDP has two main limitations: (1) in posterior sampling, it lacks generative ability and only per-

Table 5: Improving Diffusion models via posterior sampling with LDP on real-world benchmarks.

| Datasets | Scale | Metrics | LDM | +LDP | StableSR | +LDP | ResShift | +LDP | UPSR | +LDP |
|---|---|---|---|---|---|---|---|---|---|---|
| RealSR | ×4 | NIQE↓ | **6.651** | 6.830 (+0.179) | 5.948 | **5.636** (-0.312) | **8.021** | 8.027 (+0.006) | 4.854 | **4.834** (-0.020) |
| | ×4 | MANIQA↑ | **0.2904** | 0.2810 (-0.0094) | 0.3552 | **0.3644** (+0.0092) | **0.3487** | 0.3486 (-0.0001) | 0.3901 | **0.3908** (+0.0007) |
| | ×4 | CLIPIQA↑ | **0.4564** | 0.4319 (-0.0245) | 0.4840 | **0.5031** (+0.0191) | 0.5353 | **0.5354** (+0.0001) | 0.5278 | **0.5361** (+0.0083) |
| | ×4 | MUSIQ↑ | **52.09** | 50.37 (-1.72) | 55.11 | **56.56** (+1.45) | 56.85 | 56.85 | **64.82** | 64.70 (-0.12) |
| | ×4 | QAlign↑ | **2.685** | 2.610 (-0.075) | 2.607 | **2.716** (+0.109) | 3.036 | 3.036 | 3.218 | **3.231** (+0.013) |
| DPED | ×4 | NIQE↓ | **8.724** | 8.770 (+0.046) | 6.456 | **6.267** (-0.189) | 9.429 | **9.415** (-0.014) | **6.266** | 6.281 (+0.015) |
| | ×4 | MANIQA↑ | 0.2381 | **0.2418** (+0.0037) | 0.3255 | **0.3341** (+0.0086) | **0.3107** | 0.3104 (-0.0003) | 0.3151 | **0.3163** (+0.0012) |
| | ×4 | CLIPIQA↑ | **0.3718** | 0.3681 (-0.0037) | 0.4041 | **0.4053** (+0.0012) | 0.4875 | **0.4879** (+0.0004) | **0.4094** | 0.4026 (-0.0068) |
| | ×4 | MUSIQ↑ | **32.92** | 32.55 (-0.37) | 45.55 | **49.25** (+3.70) | **44.63** | 44.59 (-0.04) | 46.47 | **46.52** (+0.05) |
| | ×4 | QAlign↑ | 1.901 | **1.917** (+0.016) | 2.302 | **2.343** (+0.041) | 2.422 | **2.423** (+0.001) | **2.271** | 2.257 (-0.014) |
| RealSRSet | ×4 | NIQE↓ | 6.349 | **6.258** (-0.091) | 4.898 | **4.687** (-0.211) | **6.979** | 7.011 (+0.032) | **4.864** | 4.878 (+0.014) |
| | ×4 | MANIQA↑ | 0.3407 | **0.3470** (+0.0063) | 0.4411 | **0.4573** (+0.0162) | 0.4004 | 0.4004 | 0.4647 | **0.4720** (+0.0073) |
| | ×4 | CLIPIQA↑ | **0.5439** | 0.5311 (-0.0128) | 0.6384 | **0.6584** (+0.0200) | 0.6656 | **0.6658** (+0.0002) | 0.6709 | **0.6753** (+0.0044) |
| | ×4 | MUSIQ↑ | 58.54 | **59.52** (+0.98) | 62.73 | **62.96** (+0.23) | 66.05 | **66.06** (+0.01) | 69.68 | **69.74** (+0.06) |
| | ×4 | QAlign↑ | 3.046 | **3.089** (+0.043) | **3.193** | 3.192 (-0.001) | **3.561** | 3.560 (-0.001) | **3.705** | 3.656 (-0.049) |

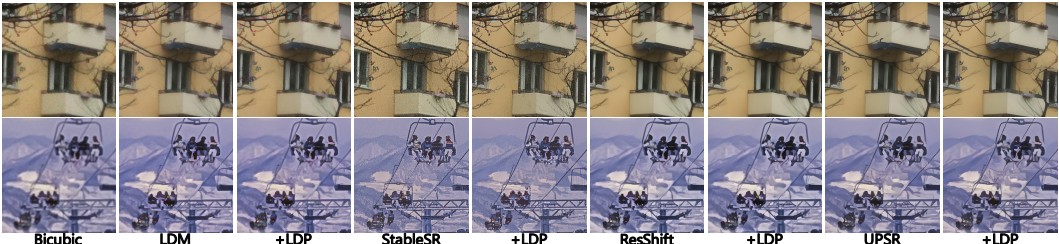

Bicubic  LDM  +LDP  StableSR  +LDP  ResShift  +LDP  UPSR  +LDP

Figure 6: Qualitative results of LDP enhances diffusion models through posterior sampling at ×4 scale SR. (**Zoom in for details**)

Table 6: Ablation study of the loss terms used in the fine-tuning stage of pretrained SwinIR models.

| Methods | $\mathcal{L}_1^{Sym}$ | $\mathcal{L}_{LPIPS}^{Sym}$ | $\mathcal{L}_{fre}^{Sym}$ | $\mathcal{L}_{fre}^{SR}$ | PSNR↑ | SSIM↑ | LPIPS↓ |
|---|---|---|---|---|---|---|---|
| **baseline** | × | × | × | × | 23.52 | 0.6458 | 0.3634 |
| **LDPV1** | × | × | × | ✓ | 23.99 | 0.6481 | 0.3591 |
| **LDPV2** | ✓ | ✓ | × | × | 24.08 | 0.6406 | 0.3585 |
| **LDPV3** | × | × | ✓ | × | 24.01 | 0.6404 | 0.3582 |
| **LDPV4** | ✓ | ✓ | ✓ | × | 24.13 | 0.6406 | 0.3609 |
| **LDPV5** | ✓ | ✓ | × | ✓ | 24.33 | 0.6499 | 0.3578 |
| **LDPV6** | × | × | ✓ | ✓ | 24.28 | **0.6500** | 0.3580 |
| **LDPV7** | ✓ | ✓ | ✓ | ✓ | **24.35** | 0.6492 | **0.3571** |

Table 7: Ablation study of the $\tau$ weight.

| $tau$ | PSNR↑ | SSIM↑ | LPIPS↓ |
|---|---|---|---|
| - | 23.52 | 0.6458 | 0.3634 |
| 0.1 | 24.15 | **0.6547** | 0.3601 |
| 1 | 24.27 | **0.6547** | 0.3595 |
| 10 | 24.30 | 0.6500 | 0.3596 |
| 100 | **24.35** | 0.6492 | **0.3571** |

forms texture rectification; (2) It does not support unpaired degradation modeling, as the generated LR image inevitably retains information from the input LR high-frequency components.

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

## A  THE USE OF LARGE LANGUAGE MODELS

We used Large Language Models to assist or polish the writing, without involving our experiments, figures, or other core contributions.

## B  ANONYMIZED LINK TO OUR CODE

Our code is available on an anonymous link for open-source access https://anonymous.4open.science/r/LDP-3CAC/.

## C  CREATION OF SYNTHETIC TESTING DATASETS

We adopt the `bsrgan_plus` degradation model Zhang et al. (2021a); Wang et al. (2021) to construct synthetic multi-degradation datasets from the DIV2K validation set. Specifically, the full `bsrgan_plus` pipeline is used to generate the hybrid degradation dataset, while four individual datasets (Downsample, Blur, Noise, and JPEG) are created by applying only the corresponding components of `bsrgan_plus`.

**Downsample.** For the downsample mode, four types of interpolation methods are employed: $D^s_{nearest}$, $D^s_{bilinear}$, $D^s_{bicubic}$ and $D^s_{down-up}$, where $s$ is the scale factor. For the $D^s_{nearest}$ method,

there is a probability that a centered $21 \times 21$ isotropic Gaussian kernel is shifted by $0.5 \times (s-1)$ pixels using a 2D linear grid interpolation technique. This step is taken to correct a potential misalignment of $0.5 \times (s-1)$ pixels towards the upper-left corner that may occur during the downsampling process. In the $D_{down-up}^{s} = D_{down}^{s/a} D_{up}^{a}$, the HR image is first downsampled by a scale factor of $s/a$ and then upsampled by a scale factor of $a$. The interpolation methods for both downsampling and upsampling are randomly selected from nearest neighbor, bilinear, or bicubic interpolation. Additionally, with a probability of 0.25, the HR image is initially resized to half of its original dimensions using a randomly selected interpolation technique. Following this resizing, $s$ is set to $s/2$ for the subsequent downsampling operation.

**Noise.** For the noise mode, a shuffle order of 5 operations is generated. These operations include: (1) Gaussian noise with a standard deviation in [2, 25], including grayscale, multivariate, and color variants with probabilities of 0.4, 0.2, and 0.4, respectively; (2) Speckle noise, applied multiplicatively with the same probability setting as Gaussian noise; (3) Poisson noise, added either globally or in grayscale with equal probability after scaling and rounding; (4) JPEG compression with a random quality factor in [30, 95]; and (5) downsampling by a factor of $s$ using a randomly selected interpolation method (nearest, bilinear, or bicubic).

**Blur.** For the blur mode, a random sequence of two operations is applied: (1) blurring the image twice using randomly generated kernels with scale factor $s$, with a 50% chance of selecting an anisotropic Gaussian kernel , and otherwise using an isotropic Gaussian kernel, with kernel size and width also randomized; and (2) downsampling by a factor of $s$ using a randomly chosen interpolation method (nearest-neighbor, bilinear, or bicubic).

**JPEG.** For the JPEG mode, a random sequence of two operations is applied: (1) simulating JPEG compression artifacts by converting the image to `uint8` format, compressing it using a randomly sampled quality factor between 30 and 95, and then decompressing it; and (2) downsampling the image by a factor of $s$ using a randomly selected interpolation method (nearest-neighbor, bilinear, or bicubic).

# D    DETAILS OF FINE-TUNING PRETRAINED SUPER-RESOLUTION MODELS

All pretrained SR models were obtained from their respective official GitHub repositories. Fine-tuning was performed using the DF2K dataset, which combines DIV2K Agustsson & Timofte (2017) and Flickr2K Lim et al. (2017), with BSRGAN Zhang et al. (2021a) employed as the degradation model. In this setting, LDP is applied only during the fine-tuning stage and is not used at inference.

**FeMaSR.** We directly fine-tuned the second-stage model of FeMaSR using its original loss functions: L1 loss, LPIPS loss, GAN loss, and a codebook-specific loss. In addition, we incorporated the frequency loss $\mathcal{L}_{fre}$ (Equ. 14) and the fine-tuning symmetry loss $\mathcal{L}_{sym}^{FT}$ (Equ. 16). The hyperparameters were set as follows: $\lambda_{fre} = 1$, $\lambda_1 = \lambda_2 = \lambda_3 = 0.1$, and $\tau = 1$. The model was fine-tuned for 100,000 iterations. Notably, even a brief fine-tuning of 1,000 iterations significantly reduces GAN-induced artifacts. Longer training durations allow the discriminator to better converge, thereby enhancing the generation of realistic and detailed textures. Experimental results demonstrate that when employing GAN loss, extended fine-tuning is typically necessary to ensure stable convergence of the discriminator.

**StableSR.** To fine-tune StableSR, we follow the original loss settings with two additional loss terms: the frequency loss $\mathcal{L}_{fre}$ (Equ. 14) and the fine-tuning symmetry loss $\mathcal{L}_{sym}^{FT}$ (Equ. 16). Since StableSR is a latent diffusion model, it is necessary to use the decoder to transform the latent features back into the RGB space. Specifically, at each diffusion step, based on DDPM Ho et al. (2020) or DDIM Song et al. (2021), the model predicts the clean image $\hat{x}_0$ from the noisy input. We first apply the decoder to convert $\hat{x}_0$ into a RGB image $X'$, which is then used to compute the frequency loss $\mathcal{L}_{fre}(X', x)$ for frequency modulation. Subsequently, $X'$ along with the high-frequency component of LR $y_{hf}$ is fed into our LDP module to generate a predicted LR image $y'$. We then apply the symmetry loss $\mathcal{L}_{sym}^{FT}(y', y)$ to further guide the super-resolution process. The hyper-parameters are set $\lambda_{fre} = 0.1$, $\lambda_1 = \lambda_2 = \lambda_3 = 0.1$ and $\tau = 1$. The model was fine-tuned for 2,000 iterations. The inference code is the same as the original StableSR with the DDPM step set as 200.

**SwinIR and MambaIR.** To fine-tune SwinIR and MambaIR, we use $\mathcal{L}_1$, $\mathcal{L}_{LPIPS}$ and $\mathcal{L}_{fre}$ (Equ. 14) to constrain HR and SR result, while use $\mathcal{L}_{sym}^{FT}$ to constrain LR and the predicted LR from our LDP. The hyper-parameters are set $\lambda_{fre} = 10$, $\lambda_1 = \lambda_2 = \lambda_3 = 1$ and $\tau = 100$. he models were fine-tuned for 1,000 iterations.

## E  DIFFUSION POSTERIOR SAMPLING WITH LDP

We evaluated how LDP enhances pre-trained diffusion models through posterior sampling, including LDM Rombach et al. (2022), StableSR Wang et al. (2024), ResShift Yue et al. (2025), and UPSR Zhang et al. (2025). Posterior sampling, as formulated in Eq. 17, is carried out without any fine-tuning. In this setting, quantitative metrics may show limited improvement. However, visual results demonstrate a notable reduction in artifacts and enhanced fidelity in the outputs of the diffusion models. For all four baseline models, the LDP parameters are set to $\tau = 100$ and $\lambda_1 = \lambda_2 = \lambda_3 = 1$. As all selected models are latent diffusion model, we should first use the Decoder to transfor the latent feature back to the color space. Specifically, for every diffusion step, according DDPM Ho et al. (2020) or DDIM Song et al. (2021), the model will get the predicted clean image $\hat{x}_0$ from the model output. We decode $\hat{x}_0$ into the RGB image $X'$, which is then combined with the high-frequency component of LR images $y_{hf}$ and passed into our LDP module to generate a predicted LR image $y'$. The fine-tuning symmetry loss $\mathcal{L}_{sym}^{FT}(y', y)$ is subsequently applied to further guide the super-resolution model.

**LDM.** We use the SR version of LDM with 50 DDIM steps, we apply LDP only every 5 steps during the last 25 steps of the sampling process. This is because LDM has already undergone super-resolution training, so the predicted clean image $\hat{x}_0$ in the early stages of the DDPM process are sufficiently close to the LR input. However, as the diffusion process progresses, the generated SR images may gradually diverge from the LR features, thereby necessitating additional guidance. Moreover, applying the DPS operation increases inference time. While applying it at every step could further improve the fidelity of the generated results, the computational overhead becomes prohibitive.

**StableSR.** We found that the SR result of StableSR exhibits a noticeable repeat-spot artifact, as illustrated in Fig. 4 and Fig. 7. We note that the artifact can be removed by subtracting noise during inference Bansal et al. (2023), a technique compatible with the inference process of StableSR. However, in our experiments, this artifact removal method was applied only in the posterior sampling setting and not during inference with fine-tuned models. Specifically, we set $P(x, t)$ as the noise diffusion process at time $t$. In each denoising step, the update can be formulated as:

$$x_{t-1} = x_{t-1} - \lambda * (\mathbf{P}(\hat{x}_0, t) + \mathbf{P}(\hat{x}_0, t-1)), \tag{18}$$

where we set $\lambda = 0.01$. We adopt 200 DDPM steps, but our LDP are applied in the last 100 steps, and only every 10 steps. For the same reasons as in LDM. We observe that applying LDP directly to StableSR without this technique tends to exacerbate the repeat-spot artifact. In contrast, applying the artifact removal prior to LDP further enhances StableSR's performance. We hypothesize that this is because StableSR possesses strong generative capability, producing super-resolved images that deviate from the LR input. Consequently, when LDP is used to enforce consistency between the SR and LR images, it may inadvertently suppress the model's generative ability.

**ResShift.** We adopt the journal version of ResShift, requiring only four steps to generate SR results, with LDP applied at each step.

**UPSR.** UPSR generates SR results in only five steps, with LDP applied at each step.

## F  EXTENDED ABLATION STUDY

**Ablation of the Patch Sise in Noise Addition Module.** Table 8 presents an ablation study investigating the effect of patch size in the patch-wise noise addition process of diffusion. We systematically vary the patch size in 1, 4, 8, 16 and evaluate each configuration on the fine-tuning pretrained SwinIR model (baseline) using the synthetic Hybrid dataset. Experimental results demonstrate that any patch configuration surpasses the baseline. When the patch size equals one, it implies that uniform noise is added across the entire image. Since a patch size of 16 attains the highest PSNR and the lowest LPIPS, we set $P = 16$ in our LDP.

Table 8: Ablation study of the Patch Size of LDP.

| Methods | $patch$ | PSNR↑ | SSIM↑ | LPIPS↓ |
|---------|---------|-------|-------|--------|
| **baseline** | - | 23.52 | 0.6458 | 0.3634 |
| **LDPp2** | 1 | 24.43 | 0.6505 | 0.3567 |
| **LDPp4** | 4 | 24.45 | 0.6519 | 0.3567 |
| **LDPp8** | 8 | 24.34 | **0.6520** | 0.3572 |
| **LDPp16** | 16 | **24.46** | 0.6513 | **0.3566** |

Table 9: Ablation study of the frequency band used in $\mathcal{L}_{sym}^{FT}$.

| Methods | $\mathbf{DWT_{fre}}$ | PSNR↑ | SSIM↑ | LPIPS↓ |
|---------|---------|-------|-------|--------|
| **baseline** | baseline | 23.52 | 0.6458 | 0.3634 |
| **LDP$_{LF}$** | LL | **24.35** | 0.6472 | 0.3573 |
| **LDP$_{HF}$** | LH+HL+HH | **24.35** | **0.6492** | **0.3571** |
| **LDP$_{ALL}$** | ALL | 24.33 | 0.6430 | 0.3574 |

**Ablation of the Frequency Band in $\mathcal{L}_{sym}^{FT}$.** Table 9 presents an ablation study on DWT frequency-band supervision. In this experiment, the pretrained SwinIR model (baseline) is fine-tuned and evaluated on the synthetic Hybrid dataset. The variants $LDP_{LF}$, $LDP_{HF}$, and $LDP_{ALL}$ apply supervision to the LL (low-frequency), LH/HL/HH (high-frequency), and all DWT sub-bands. Both $LDP_{LF}$ and $LDP_{HF}$ improve PSNR from 23.52 to 24.35, with $LDP_{HF}$ achieving slightly higher SSIM and the lowest LPIPS. In contrast, $LDP_{ALL}$ yields comparable PSNR and LPIPS but slightly lower SSIM, suggesting that focused supervision on specific frequency bands is more effective than supervising all sub-bands indiscriminately.

Table 10: Ablation study of the scale factor in LR residual acquisition phase.

| Methods | $s'$ | PSNR↑ | SSIM↑ | LPIPS↓ |
|---------|------|-------|-------|--------|
| **baseline** | - | 23.52 | 0.6458 | 0.3634 |
| **LDPsf2** | 2 | **24.35** | 0.6492 | **0.3571** |
| **LDPsf4** | 4 | 24.31 | 0.6490 | 0.3576 |
| **LDPsf8** | 8 | 24.24 | 0.6495 | 0.3582 |
| **LDPsf16** | 16 | 24.21 | **0.6496** | 0.3585 |

**Ablation of the Scale Factor in the LR Residual Acquisition Phase.** To investigate how the scale factor $s'$ affects performance, we conduct an ablation study by varying $s'$ during the fine-tuning of a pretrained SwinIR model (baseline) using the synthetic Hybrid dataset. This factor determines the high-frequency components extracted from LR images. As listed in Tab. 10, all LDP variants outperform the baseline, with the best performance achieved at $s' = 2$. As $s'$ increases, PSNR and LPIPS consistently decline, while SSIM steadily improves. This is because larger $s'$ values introduce stronger but less reliable high-frequency components into the LDP input. These components may amplify edge-like patterns that enhance SSIM but do not faithfully reflect true HR details, thereby increasing prediction errors and perceptual inconsistencies. As a result, the quality of the supervision signal deteriorates, weakening the fine-tuning effectiveness and degrading overall SR performance. These findings highlight the importance of selecting an appropriate $s'$ to balance structural sharpness and reconstruction fidelity.

**LDP contributions to existing SR models evaluated on severely degraded test dataset.** To evaluate our method on severely degraded LR images, we regard pretrained SwinIR as baseline and test SwinIR+LDP in our main text. We still use the bsrgan_plus Zhang et al. (2021a); Wang et al. (2021) Zhang et al. (2021a); Wang et al. (2021) degradation setting, while changing the maximum length (wd2) of the Gaussian blur kernel (please refer to the bsrgan_plus code), we set $wd2 \in [8, 16, 32, 64, 484]$. By default, $wd2$ is set to 8 and applied in the synthetic Hybrid dataset. As reported in Tab. 11, the results demonstrate that our method retains strong generalization under server Gaussian blur. Because the training signal is the residual obtained by subtracting the downsample-upsample from the original LR, the residual is never zero even when a heavy blur removes substantial high-frequency content, thereby providing a distinctive cue for identifying the LR corresponding to the same HR image. Tab. 12 further reports the LR-prediction evaluation of LDP. Even under severe blur (wd2 = 484), the predicted LR remains highly consistent with its initial counterpart, achieving 26.87 dB PSNR, 0.89 SSIM, and 0.1618 LPIPS. These results underscore the robustness of our LDP.

**The computational burden of LDP when it is employed for posterior sampling.** We assess the integration of LDP into the diffusion posterior sampling Chung et al. (2023) framework built upon StableSR under the synthetic Hybrid dataset, where Eq. 18 is applied to suppress artifacts.

Table 11: Ablation study on severely Gaussian blur degraded LR images. $wd2$ is the maximum length of Gaussian blur in bsrgan_plus degradation process.

| Methods | $wd2$ | PSNR↑ | SSIM↑ | LPIPS↓ |
|---|---|---|---|---|
| **baseline** | 8 | 23.52 | 0.6458 | 0.3634 |
| **+LDP** | 8 | **24.21** | **0.6496** | **0.3585** |
| **baseline** | 16 | 22.99 | 0.6296 | 0.3974 |
| **+LDP** | 16 | **23.78** | **0.6319** | **0.3932** |
| **baseline** | 32 | 22.82 | 0.6188 | 0.3967 |
| **+LDP** | 32 | **23.76** | **0.6238** | **0.3935** |
| **baseline** | 64 | 22.27 | 0.5971 | 0.4363 |
| **+LDP** | 64 | **23.15** | **0.6023** | **0.4341** |
| **baseline** | 484 | 21.24 | 0.5740 | 0.4810 |
| **+LDP** | 484 | **22.09** | **0.5812** | **0.4759** |

Table 12: Performance of LDP in LR prediction on severely Gaussian-degraded LR images.

| $wd2$ | PSNR↑ | SSIM↑ | LPIPS↓ |
|---|---|---|---|
| 8 | 29.81 | 0.9169 | 0.1009 |
| 16 | 27.61 | 0.9123 | 0.1231 |
| 32 | 27.51 | 0.9087 | 0.1215 |
| 64 | 27.32 | 0.9053 | 0.1325 |
| 484 | 26.87 | 0.8900 | 0.1618 |

Table 13: Inference time of posterior sampling with LDP in Diffusion models and its impact on performance.

| per image (s) | baseline | LDPtV1 | LDPtV2 | LDPtV3 |
|---|---|---|---|---|
| Times | 19 | 178 | 99 | 28 |
| PSNR↑ | 19.71 | **19.90** | 19.72 | 19.72 |
| SSIM↑ | 0.3756 | **0.3848** | 0.3718 | 0.3705 |
| LPIPS↓ | 0.5118 | **0.5020** | 0.5115 | 0.5057 |

Four configurations are compared: (1) baseline: the original StableSR baseline with 200 DDPM denoising iterations; (2) LDPtV1: LDP applied at every step across all 200 iterations; (3) LDPtV2: LDP applied only during the last 100 iterations; and (4) LDPtV3: LDP applied once every ten steps within the last 100 iterations. The quantitative results are reported in Tab. 13. Applying DPS at every step significantly improves the performance of diffusion models, but incurs prohibitive inference overhead. In contrast, applying LDP once every ten steps during the final 100 iterations introduces only a modest runtime increase, while still yielding performance gains over the baseline. We emphasize that no acceleration techniques such as half-precision were used during testing. All models were run in full precision on the GPU, and additional speed-ups may be achieved with alternative strategies.

Table 14: Comparison of training cost and efficiency between the proposed LDP and other plug-in methods.

| Methods | GPU memory (MiB) | Time per Iteration (s) | PSNR↑ | SSIM↑ | LPIPS↓ |
|---|---|---|---|---|---|
| **SwinIR** | 15575 | 1.413 | 23.64 | **0.6098** | 0.4541 |
| **SwinIR+LDP** | 22405 | 2.094 | **23.96** | 0.6050 | **0.4468** |
| **SwinIR+Lway** | 200768 | 22.55 | 21.11 | 0.6024 | 0.5126 |

**Evaluating training cost and efficiency of LDP against other plug-in methods.**

We report the training cost and efficiency of incorporating LDP as a loss component of SwinIR under the synthetic Hybrid dataset, in comparison with Lway Chen et al. (2024). Since the official Lway code is not publicly available, we re-implemented it following their GitHub guidelines. Using Lway as a loss component is equivalent to the original Lway paper, where the pre-trained model is 100% fine-tuned. Three configurations are compared: (1) **SwinIR**: SwinIR trained from scratch with $L_1 + L_{fre}$; (2) **S+LDP**: SwinIR trained with $L_1 + L_{fre} + \mathcal{L}_{sym}^{FT}$. Predicted LR comes from LDP. (2) **S+Lway**: SwinIR trained with $L_1 + L_{fre} + \mathcal{L}_{sym}^{FT}$. Predicted LR comes from Lway. As reported in Tab 14, LDP increases SwinIR's training GPU memory from 15,575 MiB to 22,405 MiB, extends the per-iteration runtime from 1.413 s to 2.094 s, and consequently raises the compute cost for 50,000 iterations from 21.23 h to 31.26 h. In exchange, PSNR and LPIPS improve, and

SSIM changes marginally. In contrast, Lway does not improve model performance within the same training time and consumes even more GPU memory.

## G  EXTENDED QUALITATIVE RESULTS

More visual results of the blind SR models on both synthetic and real-world benchmarks are provided in Fig. 7 and Fig. 8, respectively. Additional qualitative results of diffusion posterior sampling are presented in Fig. 9. With the assistance of LDP, existing SR models demonstrate a clear ability to suppress artifacts, preserve LR features, and generalize better to unseen degradation types. However, this approach also reveals a limitation: for models such as FeMaSR, which treat certain artifacts as part of the texture, LDP struggles to preserve the model's original ability to generate detailed textures while removing artifacts. This highlights a trade-off between artifact suppression and texture fidelity in models that implicitly rely on artifact patterns for texture synthesis.

## H  ETHICS STATEMENT

Our work focuses on single-image super-resolution and synthetic degradation modeling using publicly available or properly licensed images. No human subjects or sensitive personal data are involved. The LDP model is intended for research and image enhancement, and we acknowledge that generative image processing can be misused. We encourage responsible use and compliance with relevant legal and ethical guidelines.

## I  REPRODUCIBILITY STATEMENT

Our code is provided in Appendix B. The training details of our proposed LDP are described in Section 4.1 of the main text. The generation process of the synthetic multi-degradation datasets is presented in Appendix C. Experimental details of fine-tuning existing SR models with LDP are given in Appendix D, while Appendix E provides the details of applying LDP for posterior sampling with pre-trained diffusion models.

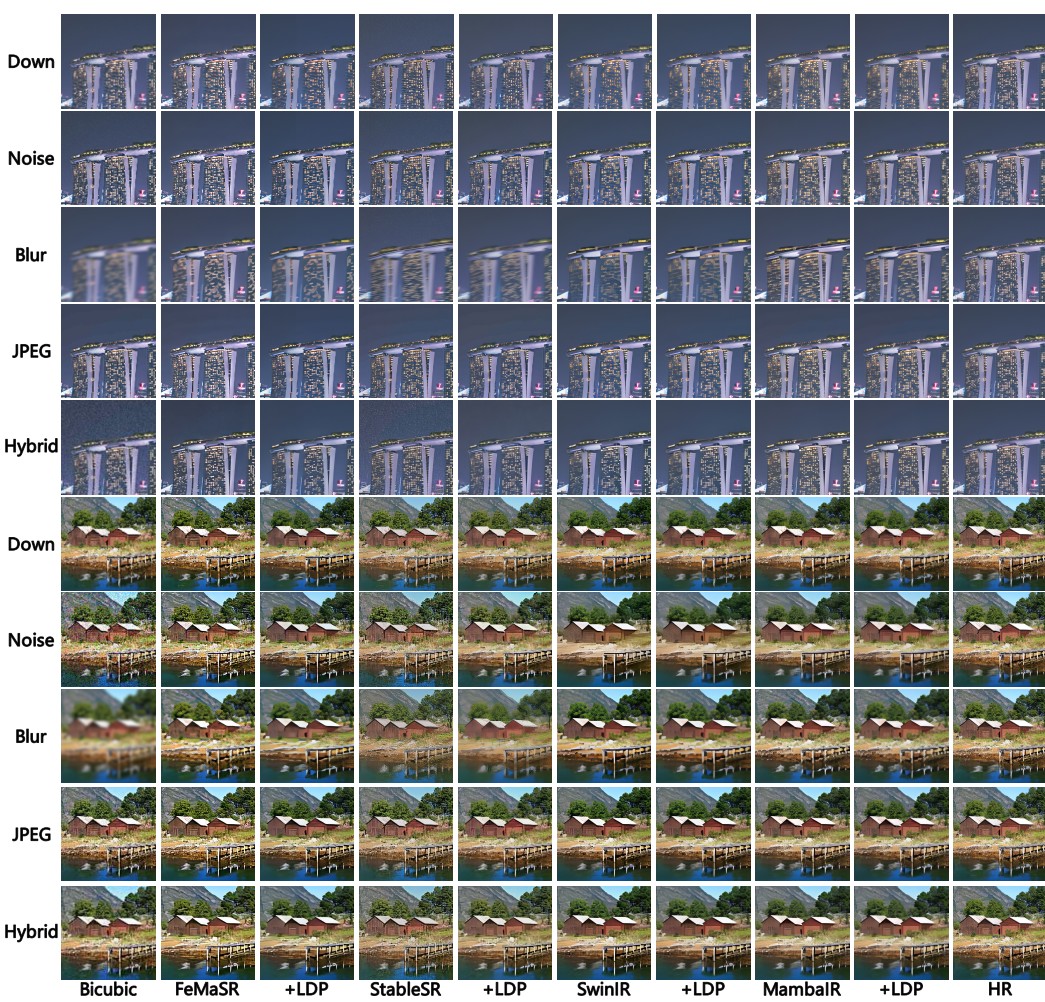

Figure 7: Qualitative results on synthetic multi-degradation datasets with $\times 4$ scale factor. (**Zoom in for details**)

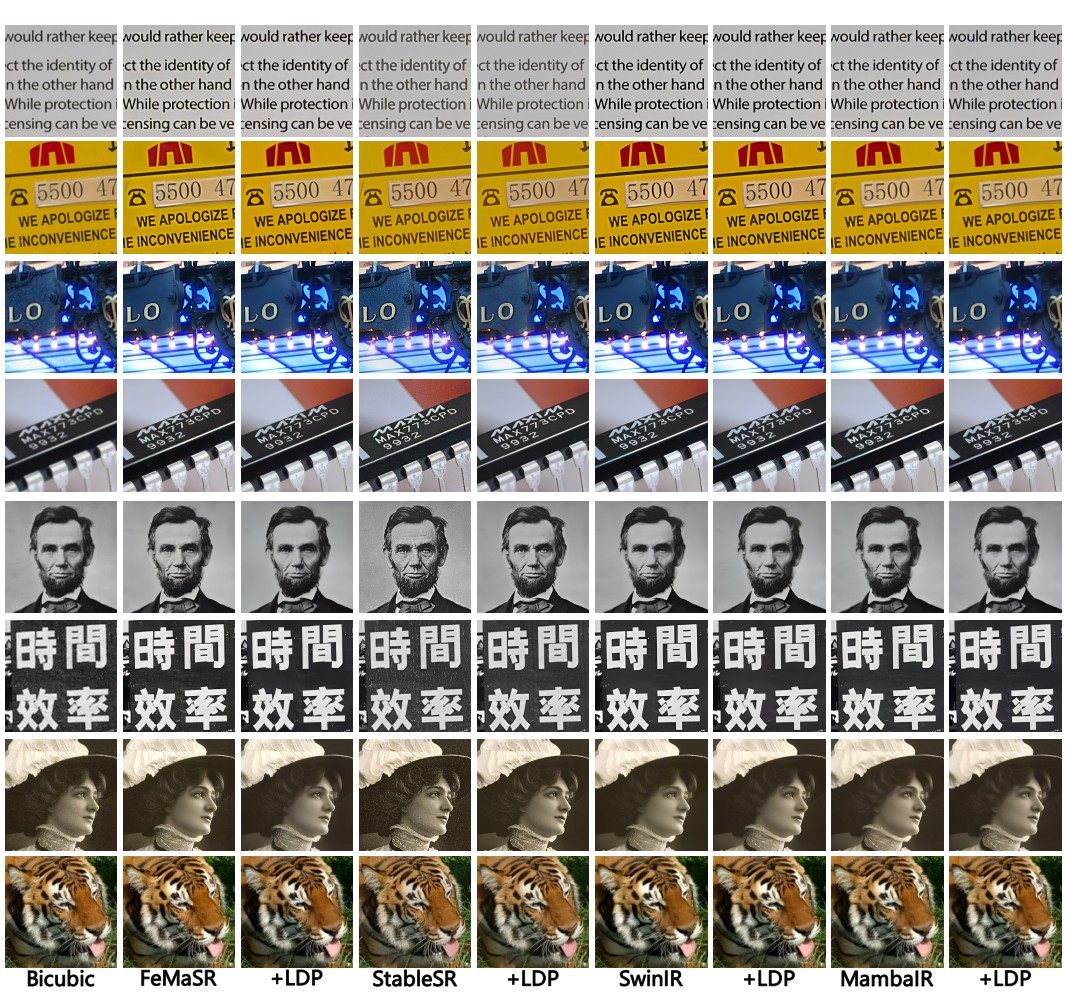

Figure 8: Qualitative results on real-world datasets with $\times 4$ scale factor. (**Zoom in for details**)

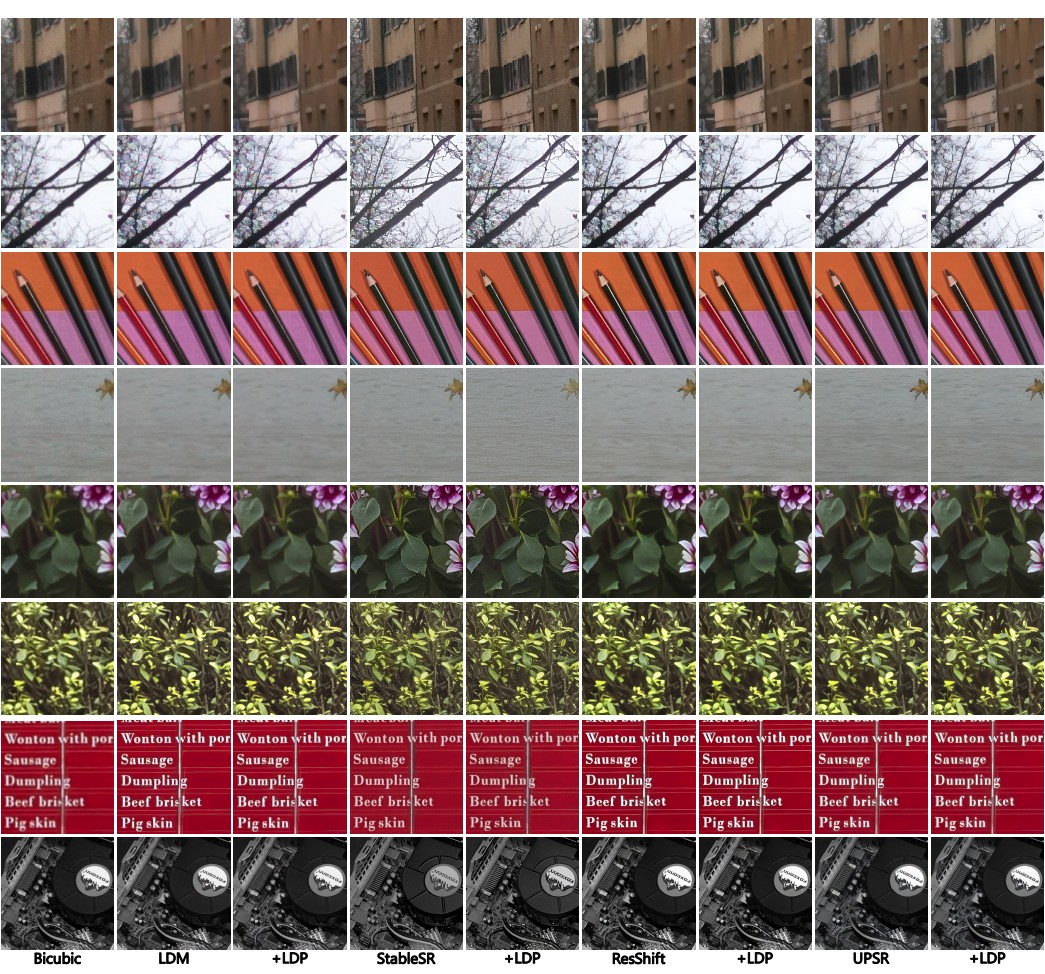

Figure 9: Qualitative results of LDP enhances diffusion models through posterior sampling at ×4 scale SR. (**Zoom in for details**)

