# OpenReview forum: "LDP: A Lightweight Denoising Plugin Enhancing Generalization in Single-Image Super-Resolution"
_ICLR.cc/2026/Conference — Submitted to ICLR 2026_

### Official Review · Reviewer_C4zd · 2025-10-30

**Soundness:** 3
**Presentation:** 3
**Contribution:** 3
**Rating:** 6
**Confidence:** 4

**Summary:**

This paper proposed a general HR to LR lightweight network, which can regularize the training and act as an auxiliary task for test-time adaptation.

**Strengths:**

1. Lightweight and plug-and-play.
2. Can be used as bot regularizer during training and corrector during test-time adaptation.
3. General to most SR models.

**Weaknesses:**

Still need paired data for training.

**Questions:**

1. I noticed that there is a cone layer before the NAM in Fig 2(a), but the input of Fig 2(c) is still images, please clarify this.
2. This LDP module is also trained on the paired data, how come it is capable of generalizing to the unseen degradation? Because noise makes HR and LR distributions comparable? Any more mathematical explanation of this?
3. When the LR is poor, I think high-freq feature of it is almost zero. How can this zero-like map help.
4. I believe there are some other HR to LR methods, can they help? Any comparison?

---

> ### Author Response · Authors · 2025-11-20
> **Official Comment by Authors (1/3)**
>
> We sincerely appreciate the time and effort you have dedicated to reviewing our manuscript and would like to address each of your concerns.
>
> >  **Q1:** *there is a conv layer before the NAM in Fig 2(a), but the input of Fig 2(c) is still images, please clarify this.*
>
> **R:** Thanks. For illustration clarity, we used the raw images directly in Fig. 2(c).
> We have revised Fig. 2 in the manuscript to prevent any potential misunderstanding.
>
> >  **Q2:** *This LDP module is also trained on the paired data, how come it is capable of generalizing to the unseen degradation?*
>
> **R:** Thanks. The theoretical foundation of LDP lies in the classical degradation model for image restoration,
> $LR = (HR \otimes k)↓_{s}+n=((HR+n) \otimes k)↓_s$，where $k$ is the degradation kernel, which we approximate using convolution kernels in LDP, and $n$ denotes noise. In principle, any degradation consistent with this classical formulation can be generalized by our LDP. Moreover, we adopt the degradation-shuffle strategy of BSRGAN [1] to generate diverse synthetic degradations during training, effectively increasing training difficulty and improving robustness. In rebuttal Table 1, which excerpts results from main text Table 1, we report LDP's performance under various degradation settings unseen during training. The numerical results show that LDP exhibits strong generalization capability.
>
> **Table 1:** Performance of LDP in LR prediction on unknown synthetic multi-degradation datasets
>
> | Method | Scale | Down | Noise | Blur | JPEG | Hybrid |
> |-|-|-|-|-|-|-|
> ||| PSNR/SSIM/LPIPS | PSNR/SSIM/LPIPS | PSNR/SSIM/LPIPS | PSNR/SSIM/LPIPS | PSNR/SSIM/LPIPS |
> |LDP |x4 | 29.15/0.9283/0.0985  | 26.71/0.8978/0.1248 | 28.41/0.9159/0.1417 |28.01/0.9243/0.0877 | 27.94/0.9173/0.1025 |
>
> Ref:
>
> [1] Kai Zhang, et al. *Designing a Practical Degradation Model for Deep Blind Image Super-Resolution*. ICCV, 2021.

---

> > ### Comment · Reviewer_C4zd · 2025-11-27
> >
> > Equation 1 has no JPEG compression, how can the model trained with it be generalized to JPEG compression?

---

> > > ### Author Response · Authors · 2025-11-27
> > > **Official Comment by Authors**
> > >
> > > Thank you for the valuable comments. Although JPEG compression is not explicitly defined in Equation 1 $ y =((x+n) \otimes k)\downarrow_s $, nor directly simulated in our LDP, JPEG artifacts such as blocking and quantization errors typically appear at the local patch level. JPEG uses 8×8 blocks, while LDP applies patch-based degradation on 16×16 regions, which still captures similar localized distortion patterns. Therefore, the degradation patterns learned through LDP enable the model to indirectly generalize to JPEG compression.
> > >
> > > In addition, the BSRGAN degradation pipeline [1] used in our training data includes JPEG-based degradation. This further improves the model’s robustness and generalization to JPEG compression through data-driven learning.
> > >
> > > [1] Kai Zhang, et al. *Designing a Practical Degradation Model for Deep Blind Image Super-Resolution*. ICCV, 2021.

---

> ### Author Response · Authors · 2025-11-20
> **Official Comment by Authors (2/3)**
>
> >  **Q3:** *When the LR is poor, I think high-freq feature of it is almost zero. How can this zero-like map help.*
>
> **R:** Thanks. The statement “When the LR is poor, the high-frequency feature of it is almost zero” holds only under certain conditions. When the LR is severely degraded, its high-frequency components relative to its HR counterpart do indeed approach zero. However, the high-/low-frequency decomposition within the LR itself still exists.
>
> In our LDP design, we extract the LR high-frequency component by subtracting its own downsample–upsample reconstruction from the original LR. Because the downsample–upsample process is lossy, this operation cannot yield a perfect zero map. As a result, the extracted $LR_{hf}$ remains informative and can serve as a condition for LDP to distinguish different LR instances corresponding to the same HR.
>
> In Appendix Table 11 (originally Appendix Table 9), we also evaluate LDP on LR images degraded by heavy blur kernels—i.e., scenarios where the high-frequency feature of LR is almost zero. The results show that LDP can still improve the SR model’s performance under such conditions. In this experiment, we still use the bsrgan\_plus  degradation (a blend of BSRGAN [1] and RealESRGAN [2]) setting, while changing the maximum length (wd2) of the Gaussian blur kernel (please refer to the bsrgan\_plus code), we set $wd2 \in [8,16,32,64,484] $. As reported in rebuttle Table 2, the results demonstrate that our method retains strong generalization when the high-freq feature of LR is almost zero. Rebuttle Table 3 further reports the LR-prediction evaluation of LDP. Even under severe blur (wd2 = 484), the predicted LR remains highly consistent with its initial counterpart. These results underscore the robustness of our LDP.
>
> **Table 2:** Ablation study on severely Gaussian blur degraded LR images. $wd2$ is the maximum length of Gaussian blur in bsrgan_plus degradation process.
>
> | Methods     | $wd2$ | PSNR ↑  | SSIM ↑  | LPIPS ↓  |
> |-|-|-|-|-|
> | **SwinIR** | 8     | 23.52   | 0.6458  | 0.3634   |
> | **+LDPSR**  | 8     | **24.21** | **0.6496** | **0.3585** |
> | **baseline** | 16    | 22.99   | 0.6296  | 0.3974   |
> | **+LDPSR**  | 16    | **23.78** | **0.6319** | **0.3932** |
> | **baseline** | 32    | 22.82   | 0.6188  | 0.3967   |
> | **+LDPSR**  | 32    | **23.76** | **0.6238** | **0.3935** |
> | **baseline** | 64    | 22.27   | 0.5971  | **0.4363** |
> | **+LDPSR**  | 64    | **23.15** | **0.6023** | **0.4341** |
> | **baseline** | 484   | 21.24   | 0.5740  | 0.4810   |
> | **+LDPSR**  | 484   | **22.09** | **0.5812** | **0.4759** |
>
>
> **Table 3:** Performance of LDP-SR in Generating LR Images on severely Gaussian-degraded LR images.
> | $wd2$ | PSNR ↑  | SSIM ↑  | LPIPS ↓  |
> |-|-|-|-|
> | 8     | 29.81   | 0.9169  | 0.1009   |
> | 16    | 27.61   | 0.9123  | 0.1231   |
> | 32    | 27.51   | 0.9087  | 0.1215   |
> | 64    | 27.32   | 0.9053  | 0.1325   |
> | 484   | 26.87   | 0.8900  | 0.1618   |
>
> Ref:
>
> [1] Kai Zhang, et al. *Designing a Practical Degradation Model for Deep Blind Image Super-Resolution*. ICCV, 2021.
>
> [2] Xintao Wang, et al. *Real-ESRGAN: Training Real-World Blind Super-Resolution with Pure Synthetic Data*. ICCVW, 2025.

---

> ### Author Response · Authors · 2025-11-20
> **Official Comment by Authors (3/3)**
>
> >  **Q4:** *I believe there are some other HR to LR methods, can they help? Any comparison?*
>
> **R:** Thanks. You are right. Our LDP is capable of handling not only blind SR but also individual tasks such as denoising, deblurring, and JPEG compression artifact reduction (JPEG CAR), as well as mixtures of these tasks. Main text Table 3 (originally main text Table 2) presents these results. We evaluate five degradation settings: Downsample, which corresponds to the standard super-resolution task; Noise, which corresponds to denoising task; Blur degradation, which corresponds to deblurring task; JPEG degradation, which corresponds to JPEG compression artifact reduction task; and Hybrid degradation, which corresponds to our blind super-resolution setting. We present the experimental results again in the form of rebuttle Table 4. The results show that our model exhibits good generalization across these tasks.
>
>
> **Table 4:** Performance improvements of blind SR models across diverse architectures using our
> proposed LDP on synthetic multi-degradation benchmarks. We generate synthetic benchmarks from
> the DIV2K validation set using five types of degradation: (1) Downsampling (Down), (2) Noise, (3)
> Blur, (4) JPEG, and (5) Hybrid degradations following bsrgan plus defaults.
>
> | Datasets | Scale | Metrics | FeMaSR | +LDP | StableSR | +LDP | SwinIR | +LDP | MambaIR | +LDP |
> |-|-|-|-|-|-|-|-|-|-|-|
> | **Down** | ×4 | PSNR↑ | 24.22 | **25.06** (+0.84) | 20.35 | **21.73** (+1.38) | 25.44 | **25.86** (+0.42) | 26.58 | **26.63** (+0.05) |
> | | ×4 | SSIM↑ | 0.6793 | **0.7105** (+0.0312) | 0.4998 | **0.5642** (+0.0644) | 0.7210 | **0.7242** (+0.0032) | 0.7393 | **0.7403** (+0.0010) |
> | | ×4 | LPIPS↓ | 0.2637 | **0.2490** (-0.0147) | 0.3746 | **0.2870** (-0.0876) | 0.2579 | **0.2538** (-0.0041) | 0.2054 | **0.2005** (-0.0049) |
> | **Noise** | ×4 | PSNR↑ | 22.82 | **23.84** (+1.02) | 19.95 | **21.48** (+1.53) | 24.34 | **25.04** (+0.70) | 26.11 | **26.34** (+0.23) |
> | | ×4 | SSIM↑ | 0.6519 | **0.6957** (+0.0438) | 0.4569 | **0.5599** (+0.1030) | 0.7130 | **0.7198** (+0.0068) | 0.7382 | **0.7411** (+0.0029) |
> | | ×4 | LPIPS↓ | 0.2788 | **0.2624** (-0.0164) | 0.4279 | **0.3040** (-0.1239) | 0.2676 | **0.2659** (-0.0017) | 0.2279 | **0.2219** (-0.0060) |
> | **Blur** | ×4 | PSNR↑ | 24.12 | **24.42** (+0.30) | 19.98 | **21.50** (+1.52) | 24.03 | **24.67** (+0.64) | 24.99 | **25.33** (+0.34) |
> | | ×4 | SSIM↑ | 0.6639 | **0.6787** (+0.0148) | 0.4373 | **0.5437** (+0.1064) | 0.6764 | **0.6833** (+0.0069) | 0.6892 | **0.6942** (+0.0050) |
> | | ×4 | LPIPS↓ | **0.3168** | 0.3199 (+0.0031) | 0.5112 | **0.4763** (-0.0349) | 0.3197 | **0.3168** (-0.0029) | 0.2768 | **0.2751** (-0.0017) |
> | **JPEG** | ×4 | PSNR↑ | 22.92 | **23.87** (+0.95) | 20.17 | **21.91** (+1.74) | 24.55 | **25.27** (+0.72) | 26.36 | **26.59** (+0.23) |
> | | ×4 | SSIM↑ | 0.6696 | **0.7068** (+0.0372) | 0.5141 | **0.5943** (+0.0802) | 0.7301 | **0.7372** (+0.0071) | 0.7497 | **0.7538** (+0.0041) |
> | | ×4 | LPIPS↓ | 0.2633 | **0.2508** (-0.0125) | 0.3682 | **0.2767** (-0.0915) | 0.2535 | **0.2506** (-0.0029) | 0.2113 | **0.2063** (-0.0050) |
> | **Hybrid** | ×4 | PSNR↑ | 23.40 | **23.72** (+0.32) | 19.27 | **21.43** (+2.16) | 23.52 | **24.35** (+0.83) | 24.35 | **24.71** (+0.36) |
> | | ×4 | SSIM↑ | 0.6211 | **0.6392** (+0.0181) | 0.3656 | **0.5197** (+0.1541) | 0.6458 | **0.6492** (+0.0034) | 0.6587 | **0.6636** (+0.0049) |
> | | ×4 | LPIPS↓ | **0.3453** | 0.3516 (+0.0063) | 0.5727 | **0.4461** (-0.1266) | 0.3634 | **0.3571** (-0.0063) | 0.3244 | **0.3210** (-0.0034) |

---

> ### Author Response · Authors · 2025-11-27
> **Looking forward to discussion**
>
> We sincerely appreciate the time and effort you have invested in reviewing our submission. We have carefully addressed your comments and made corresponding revisions to the manuscript. In addition, we have incorporated feedback from the other reviewers, which we hope will help resolve any further questions you may have.
>
> As the discussion period is ending soon, we wanted to follow up and would be grateful for any additional feedback or concerns you might wish to share, so that we can address them promptly.
>
> Thank you once again for your valuable time and dedication to the review process.

---

### Official Review · Reviewer_hH62 · 2025-11-02

**Soundness:** 3
**Presentation:** 3
**Contribution:** 3
**Rating:** 6
**Confidence:** 4

**Summary:**

The paper proposes LDP, a lightweight denoising autoencoder plug-in designed to improve the generalization of single-image super-resolution (SISR) models to unseen degradations. The key idea is to predict the LR image from the SR/HR pathway and enforce cyclic LR consistency during training of any SR backbone. At test time, the same learned degradation model can guide diffusion SR via Diffusion Posterior Sampling (DPS) to reduce artifacts. Concretely, LDP corrupts HR patches with independent noise, uses a small CNN denoiser that approximates blur kernels, and aligns HR and LR features after noise injection, yielding a symmetric LR loss in the Fourier domain and a DPS guidance term that nudges samples toward LR-consistent solutions. The method is plug-and-play, can be used either as a training loss or an inference post-processing step, and shows consistent gains across diverse SR models in both synthetic and real-world settings.

**Strengths:**

The paper proposes a practical and coherent contribution that is original in its combination of ideas: a tiny conditional denoiser that learns HR to LR degradation to (i) enforce LR-cycle and frequency consistency during training and (ii) guide diffusion SR via DPS at inference, yielding a unified degradation prior across regimes. The design is simple, data-efficient, and plug-and-play. Experiments span multiple SR backbones and degradation types, yielding consistent gains, and ablations isolate the effects of key components (symmetric LR loss, frequency term, DPS). The figures and algorithms are easy to follow, limitations are stated, and implementation details (params, schedules, compute) support reproducibility. Significance is strong for the SR community: the module is a lightweight, architecture-agnostic, and improves robustness to unseen degradations with minimal engineering overhead.

**Weaknesses:**

The paper would benefit from direct, same-protocol comparisons to earlier degradation-modeling methods (e.g., DRN, DualSR, SCL-SASR) to isolate LDP’s incremental value. As written, the related-work framing is stronger than the empirical head-to-head evidence.  Real-world results are mixed; on RealSR, some non-reference metrics worsen for certain backbones, so adding broader real-image datasets or RAW pipelines would better substantiate generalization claims.   Diffusion Posterior Sampling brings substantial runtime overhead with modest quantitative gains unless applied very frequently. Reporting FLOPs and latency, along with a speed-quality, would clarify deployability.    The guidance also interacts poorly with StableSR (a repeat-spot artifact) without an additional mitigation step, suggesting the need for a deeper analysis of when guidance helps or harms strong generative priors.  Evidence is concentrated at ×4 with limited sensitivity analysis. Multi-scale (×2/×3) results and parameter sweeps would improve external validity.    Finally, while LDP is lightweight, the training overhead is not fully characterized across backbones, and summarizing memory and performing an ablation of LDP capacity vs. benefit would make costs transparent.

**Questions:**

Add an experiment against DRN, DualSR, and SCL-SASR under the same data/degradation protocols to isolate LDP’s advantage over prior degradation-modeling pipelines? This would strengthen Section 2.2’s positioning with empirical evidence.

Real-world results on RealSR/DPED/RealSRSet are promising but mixed for some models/metrics. Can you expand the real-image evaluation (e.g., additional datasets or RAW pipelines) and explain when LDP helps or hurts no-reference metrics like CLIPIQA/NIQE?

Please report complete runtime/compute overhead for DPS guidance across models (wall-clock per image, FLOPs, peak memory), and include a speed–quality Pareto curve. Table 11 suggests significant slowdowns when guidance is frequent—how does this trade-off look across step counts and models?

For StableSR, LDP interacts with the repeat-spot artifact and requires a noise-subtraction mitigation. Can you analyze why the interaction occurs, quantify the extent to which the mitigation helps, and clarify whether similar interactions occur in LDM/ResShift/UPSR?

Hyperparameters are largely fixed (e.g., τ=100; loss weights). Could you provide sensitivity curves for τ and the frequency-loss weight and discuss recommended defaults per backbone? An expanded ablation would help practitioners robustly tune LDP.

Most results emphasize ×4 SR. Can you report multi-scale (×2/×3/×4) performance and discuss whether the LR-HF conditioning or noise schedule needs scale-specific adjustments?

Please characterize training overhead more fully across backbones (total wall-clock for fine-tuning, GPU type, batch sizes) and, if possible, include an ablation of LDP capacity (e.g., parameter count or depth) vs. benefit. Table 12 is a good start, but it focuses on SwinIR.

In the Limitations, note that LDP is not generative in posterior sampling and relies on LR high-frequency conditioning. Could discuss scenarios where this reliance risks information leakage or trivial solutions, and propose safeguards or diagnostics?

---

> ### Author Response · Authors · 2025-11-20
> **Official Comment by Authors (1/5)**
>
> We sincerely appreciate the time and effort you have dedicated to reviewing our manuscript and would like to address each of your concerns.
>
> >  **Q1:** *Add an experiment against DRN, DualSR, and SCL-SASR under the same data/degradation protocols*
>
> **R:** Thanks. To demonstrate the superiority of our model under various degradation conditions, we provide a comparison between LDP and other degradation models on LR prediction. We compare against DRN [1] and DualSR [2]. DRN [1] adds a degradation branch that projects SR outputs back to the LR domain to enforce reconstruction consistency and improve stability. DualSR [2] introduces a dual-path framework in which a GAN-based downsampler and an upsampler are jointly trained with cycle consistency to model and reverse image-specific degradations.
>
> In this experiment, we use the same datasets as in main text Table 1. Five types of synthetic degradations are included: (1) downsampling, (2) noise, (3) blur, (4) JPEG compression, and (5) hybrid degradations following the BSRGAN-plus defaults. We first generate SR images using SwinIR, and then apply the degradation models provided by DRN, DualSR and LDP to obtain the predicted LR images from the SR outputs. These predictions are then compared with the LR inputs to the SR model, and the results are reported in rebuttle Table 1. In addition, rebuttle Table 2 reports the similarity between the LR images produced by the degradation models and the downsampled SR images produced by SwinIR. A higher similarity indicates that the degradation model fails to apply the specific degradations implied by the input LR and instead behaves like a simple downsampler.
>
> As shown in the tables, our LDP performs consistently well across different degradation types. The similarity between the LDP-generated LR and the downsampled SR is significantly lower than that between the LDP-generated LR and the input LR, demonstrating that LDP does not collapse into trivial downsampling. In contrast, DRN behaves almost identically to bicubic downsampling. Its inputs include only HR/SR images without any conditional information, so it cannot correctly map an SR image to multiple possible LR variants. Its similarity to the downsampled SR image is much higher than its similarity to the LR input. DualSR appears to struggle with handling diverse degradation types.
>
> We have revised Section 4.2 of the manuscript. The changes are temporarily highlighted in blue for ease of review.
>
> **Table 1:** Performance of multiple degradation models in LR prediction on synthetic multi-degradation datasets.
> | Method | Down | Noise | Blur | JPEG | Hybrid |
> |-|-|-|-|-|-|
> || PSNR/SSIM/LPIPS | PSNR/SSIM/LPIPS | PSNR/SSIM/LPIPS | PSNR/SSIM/LPIPS | PSNR/SSIM/LPIPS |
> |DRN |**32.05**/**0.9539**/**0.0794** |**27.25**/0.7812/0.2474 |26.38/0.8273/0.3207 |**29.65**/**0.9270**/**0.0826** |27.03/0.8098/0.3360 |
> |DualSR | 19.58/0.4814/0.1408 |18.77/0.4712/0.1399 |19.36/0.4911/0.1844 |18.57/0.4612/0.1492 | 19.36/0.4883/0.2130 |
> |LDP | 29.15/0.9283/0.0985  | 26.71/**0.8978**/**0.1248** | **28.41**/**0.9159**/**0.1417** |28.01/0.9243/0.0877 | **27.94**/**0.9173**/**0.1025** |
>
>
> **Table 2:** Similarity between the LR images generated by multiple degradation models and the downsampled SR images.
> | Method | Down | Noise | Blur | JPEG | Hybrid |
> |-|-|-|-|-|-|
> || PSNR/SSIM/LPIPS | PSNR/SSIM/LPIPS | PSNR/SSIM/LPIPS | PSNR/SSIM/LPIPS | PSNR/SSIM/LPIPS |
> |DRN |**34.02/0.9638/0.0365** |**31.57/0.9590/0.0436** |**34.99/0.9692/0.0306** |**31.35/0.9587/0.0467** |**35.10/0.9679/0.0296** |
> |DualSR |22.58/0.6689/0.1264 |20.79/0.6502/0.1040 |22.57/0.7044/0.1262 |20.46/0.6356/0.1279 |22.85/0.7164/0.1175 |
> |LDP |28.41/0.8895/0.1551 |25.93/0.7508/0.3043 |25.04/0.7596/0.3278 |27.42/0.8886/0.1293 |26.28/0.7597/0.3586 |
>
> Ref:
>
> [1] Yong Guo, et al.Closed-loop matters: Dual regression networks for single image super-resolution. CVPP, 5406–5415, 2020.
>
> [2] Mohammad Emad, el al. DualSR: Zero-shot dual learning for real-world super-resolution. In IEEE Winter Conference on Applications of Computer Vision, 1629–1638, 2021.

---

> ### Author Response · Authors · 2025-11-20
> **Official Comment by Authors (2/5)**
>
> >  **Q2:** *Can you expand the real-image evaluation (e.g., additional datasets or RAW pipelines) and explain when LDP helps or hurts no-reference metrics like CLIPIQA/NIQE?*
>
> **R:** Thanks. We extended the experiments from main text Table 4 (originally main text Table 3) by including results on two additional real-world datasets, RealDeg [1] and RealPhoto60 [2]. RealDeg contains 238 images, including old photographs, classic film stills, and social media images, to evaluate our method across diverse degradation types. RealPhoto60 contains 60 real-world low-quality images collected from RealSR, DRealSR, Real47, and online sources, featuring a wide variety of content such as animals, plants, faces, buildings, and landscapes.
>
> The experimental results are shown in rebuttle Table 3. We did not report MUSIQ metrics for RealDeg because the images in this dataset are too large for our available computational resources to process this metric. Overall, the quantitative trends are consistent with those reported in main text Table 4 (originally main text Table 3). After incorporating LDP, the numerical results for FeMaSR slightly decrease on most metrics, but visual inspection shows that LDP significantly reduces visible artifacts. For the other three models, incorporating LDP improves the numerical performance across metrics. This suggests that changes in no-reference metrics can depend on the characteristics of the SR model. For example, FeMaSR exhibits noticeable artifacts, particularly at edges, with over-enhanced textures that current no-reference evaluation models often cannot distinguish from legitimate textures. Our LDP corrects such artifacts by enforcing consistency with the LR input, but a side effect is a slight reduction in the model’s generative ability.
>
>
> **Table 3:** Performance improvements of blind SR models across diverse architectures using our proposed LDP on real-world benchmarks.
>
> |Datasets|Scale|Metrics|FeMaSR|+LDP|StableSR|+LDP|SwinIR|+LDP|MambaIR|+LDP|
> |-|-|-|-|-|-|-|-|-|-|-|
> |RealSR|×4|NIQEL|**4.708**|5.533|7.446|**6.331**|4.773|**4.838**|**5.330**|5.350|
> ||×4|MANIQA↑|0.3430|**0.3654**|0.3303|**0.3548**|0.3510|**0.3742**|0.2882|**0.3374**|
> ||×4|CLIPIQA↑|**0.5645**|0.4482|0.4886|**0.5213**|0.4739|**0.5478**|0.3989|**0.4642**|
> ||×4|MUSIQ↑|58.94|**60.70**|52.99|**59.26**|59.67|**61.91**|51.87|**57.85**|
> ||×4|QAlign↑|3.695|**3.860**|2.347|**2.646**|3.820|**3.877**|3.631|**3.766**|
> |DPED|×4|NIQEL|**5.045**|5.704|7.616|**7.228**|4.982|**4.821**|5.983|**5.430**|
> ||×4|MANIQA↑|**0.3102**|0.2719|**0.3056**|0.2970|0.2637|**0.2832**|0.2334|**0.2767**|
> ||×4|CLIPIQA↑|**0.5570**|0.3610|**0.3968**|0.3843|0.3402|**0.4538**|0.3083|**0.3850**|
> ||×4|MUSIQ↑|**49.14**|44.07|42.97|**45.08**|42.10|**45.91**|35.25|**44.64**|
> ||×4|QAlign↑|**3.429** | 3.262 | 2.033 | **2.311** | 2.988 | **3.090** | 3.192 | **3.248** |
> | RealSRSet | ×4 | NIQEL | **5.236** | 5.952 | 6.090 | **5.586** | **5.424** | 5.441 | **5.726** | 5.893 |
> |  | ×4 | MANIQA↑ | **0.4006** | 0.4002 | 0.3904 | **0.4012** | 0.3740 | **0.3938** | 0.2978 | **0.3555** |
> |  | ×4 | CLIPIQA↑ | **0.6874** | 0.5683 | 0.6057 | **0.6214** | 0.5843 | **0.6376** | 0.4793 | **0.5428** |
> |  | ×4 | MUSIQ↑ | **64.65** | 64.07 | 60.15 | **62.84** | 63.60 | **65.33** | 55.96 | **61.28** |
> |  | ×4 | QAlign↑ | 3.776 | **3.870** | 2.916 | **3.247** | 2.749 | **3.322** | 3.434 | **3.632** |
> | RealDeg | ×4 | NIQE↓ |**3.813**  |4.549  |7.227  |**6.532** | **3.899**  |3.921  |4.717  |**4.430**  |
> |  | ×4 | MANIQA↑ |0.3017  |**0.3232**  | **0.3126** |0.2671 | 0.3170  |**0.3284**  |0.2778  |**0.3014**  |
> |  | ×4 | CLIPIQA↑ |**0.6001**  |0.4727  |0.4502  |**0.4750** |0.4951  |**0.5567**  |0.4165  |**0.4765**  |
> |  | ×4 | QAlign↑ |4.182  |**4.231**  |2.038  |**2.388** |4.221  |**4.229**  |4.198  |**4.199**  |
> | RealPhoto60 | ×4 | NIQE↓ |**3.719**  |4.878  |5.382  |**4.160** |**3.418**   |3.796  |4.347  |**4.291** |
> |  | ×4 | MANIQA↑ |0.3009  |**0.3320**  |0.3250  |**0.3836** |0.3578  |**0.3592**  |0.3181  | **0.3362** |
> |  | ×4 | CLIPIQA↑ |**0.5160**  |0.4356  |0.6072 |**0.6519** |0.5206  |**0.5594**  |0.5000  |**0.5279**  |
> |  | ×4 | MUSIQ↑ |**50.27**  |47.41  |61.97  |**65.59** |49.15  |**49.78**  |45.93  |**47.21**  |
> |  | ×4 | QAlign↑ |2.027  |**2.459**  |2.774  |**3.621** |**2.893**  |2.685  |**3.278**  | 3.151 |
>
> Ref:
>
> [1] Junyang Chen, et al. FaithDiff: Unleashing Diffusion Priors for Faithful Image Super-resolution. CVPR, 2025.
>
> [2] Fanghua Yu, et al. Scaling up to excellence: Practicing model scaling for photorealistic image restoration in the wild. CVPR, 2024.

---

> ### Author Response · Authors · 2025-11-20
> **Official Comment by Authors (3/5)**
>
> >  **Q3:** *Please report complete runtime/compute overhead for DPS guidance across models*
>
> **R:** Thanks. We evaluate the runtime and computational overhead of applying LDP for Diffusion Posterior Sampling, using the four diffusion models LDM, StableSR, ResShift, and UPSR introduced in Appendix Section E. The results are summarized in rebuttle Table 4. Our testing protocol is consistent with Appendix Table 13 (originally Appendix Table 11), conducted on the synthetic Hybrid dataset. The application of LDP is tailored to each model's sampling steps: for LDM (50 DDIM steps), it is integrated every 5 steps during the last 25 steps; for StableSR (200 DDPM steps), it is activated every 10 steps over the final 100 steps; while for ResShift (4 steps) and UPSR (5 steps), it is employed at every step. We believe that this trade-off has largely diminished as diffusion-based image restoration models have advanced. The number of required iterations has been significantly reduced in recent methods, with state-of-the-art models typically relying on fewer than ten steps; for instance, ResShift uses four steps and UPSR uses five steps. Applying LDP at each step in such settings adds only minimal runtime while still improving results. For diffusion models with many more iterations, one may apply LDP at every step to obtain optimal performance; to merely suppress prominent artifacts, a sparse application (e.g., every ten steps in later iterations) is sufficient, as it adds minimal runtime while improving results.
>
> **Table 4:** Inference time of posterior sampling with LDP in Diffusion models and its impact on performance.
>
> | Methods      | GPU memory (MiB) | Time per Image (s) | PSNR↑ | SSIM↑ | LPIPS↓ |
> |-|-|-|-|-|-|
> | LDM | 3217 |1.24 |22.49 | 0.5713 | **0.349**3 |
> | +LDP | 15093 | 3.37 | **22.51** |**0.5722** |0.3490 |
> | StableSR |15603  |19.84 | 19.71  | **0.3756** | 0.5118 |
> | +LDP |44897 |28.03 | **19.72** | 0.3705 | **0.5057** |
> | ResShift | 17603 | 0.45 | **21.96** | **0.5379** | 0.4244 |
> | +LDP | 27143 | 2.14 | **21.96** | 0.5371 | **0.4223** |
> | UPSR | 6713 |0.58 |**22.21** |**0.5528** |0.4320 |
> | +LDP |18049 |4.94 | 22.20 | 0.5513 | **0.4312** |
>
>
> >  **Q4:** *Can you analyze why the repeat-spot artifact of StableSR occurs, quantify the extent to which the mitigation helps, and clarify whether similar interactions occur in LDM/ResShift/UPSR?*
>
> **R:** Thanks. This type of repeat-spot artifact does not occur in LDM, ResShift, or UPSR, and it is likely caused by the computational process specific to StableSR. By combining the results from main text Tables 4-5 (originally main text Tables 3-4), we can quantify the effect of the artifact removal method [1] on StableSR. The aggregated results are presented in rebuttle Table 5. For each dataset, the first row shows the performance of StableSR, the second row shows StableSR fine-tuned with LDP, the third row shows StableSR with the artifact-removal method applied directly, and the fourth row shows StableSR with the artifact-removal method followed by our LDP for posterior sampling. It can be observed that while fine-tuning StableSR on Stable Diffusion 2.1 enables it to handle multiple degradation types for super-resolution, this comes at the cost of degrading the generative capability of SD2.1 itself. In contrast, our LDP guides the generation process to produce outputs that are more consistent with the LR input, thereby resulting in superior image quality.
>
>
> **Table 5:** Comparison of the StableSR baseline, LDP-fine-tuned baseline, AMP-enhanced baseline, and LDP post-processed results.
>
> | Datasets | Scale | Methods | NIQE | MANIQA | CLIPIQA | MUSIQ | QAlign|
> |- |- |- |- |- |- |- |- |
> | RealSR| x4 | StableSR  |7.446 |0.3303 |0.4886 |52.99 |2.347|
> |  |x4 |  +LDP (FT) |6.331 |0.3548 |**0.5213** |**59.26** |2.646|
> |  |x4 | + ARM  |5.948 |0.3552 |0.4840 |55.11 |2.607|
> |  |x4 | + ARM + LDP (PS)  |**5.636** |**0.3644** |0.5031 |56.56 |**2.726**|
> |DPED | x4 | StableSR   |7.616 |0.3056 |0.3968 |42.97 |2.033|
> | | x4 |  +LDP (FT)  |7.228 |0.2970 |0.3843 |45.08 |2.311|
> | | x4 |+ ARM  |6.456 |0.3255 |0.4041 |45.55 |2.302|】
> | | x4 |+ ARM + LDP (PS)  |**6.267** |**0.3341** |**0.4053** |**49.25** |**2.343**|
> | RealSRSet | x4 | StableSR  |6.090 |0.3904 |0.6057 |60.15 |2.916|
> | | x4 |  +LDP (FT)  |5.586 |0.4012 |0.6214 |62.84 |3.247|
> | | x4 | + ARM  |4.898 |0.4411 |0.6384 |62.73 |**3.193**|
> | | x4 | + ARM + LDP (PS)  |**4.687** |**0.4573** |**0.6584** |**62.96** |3.192|
>
> Ref:
>
> [1] Arpit Bansal, et al. Cold diffusion: Inverting arbitrary image transforms without noise. NeurIPS, 2023.

---

> ### Author Response · Authors · 2025-11-20
> **Official Comment by Authors (4/5)**
>
> >  **Q5:** *Could you provide sensitivity curves for τ and the frequency-loss weight and discuss recommended defaults per backbone?*
>
> **R:** Thanks. As shown in main text Table 6 (originally main text Table 5), we evaluate the impact of different loss terms during fine-tuning using SwinIR as the baseline method. Additionally, we include results using MambaIR as the baseline (rebuttle Table 6). We also provide an ablation study on the parameter $\tau$, reported separately for MambaIR and SwinIR as baselines (rebuttle Table 7).
>
> As noted in the main text (Lines 473–474), “The LDP parameters can be universally configured as $\tau = 100$ and $\lambda_1 = \lambda_2 = \lambda_3 = 1$ for any super-resolution model, leading to improved generalization performance.” We set $\tau = 100$ primarily to improve computational efficiency. We emphasize that, when fine-tuning existing models, the most critical factor is not the weight settings, but rather the learning rate. As a rule of thumb, we recommend setting the learning rate to 0.00001.
>
> **Table 6:** Ablation study of the loss terms used in the fine-tuning stage of pretrained SwinIR and MambaIR models.
> | Methods  | $L_1^{Sym}$ | $L^{Sym}_{LPIPS}$ | $L^{Sym}_{fre}$ | $L^{SR}_{fre}$ | PSNR↑  | SSIM↑  | LPIPS↓  |
> |-|-|-|-|-|-|-|-|
> | SwinIR | ✗      | ✗           | ✗         | ✗        | 23.52  | 0.6458 | 0.3634 |
> |     | ✗      | ✗           | ✗         | ✓        | 23.99  | 0.6481 | 0.3591 |
> |    | ✓      | ✓           | ✗         | ✗        | 24.08  | 0.6406 | 0.3585 |
> |    | ✗      | ✗           | ✓         | ✗        | 24.01  | 0.6404 | 0.3582 |
> |    | ✓      | ✓           | ✓         | ✗        | 24.13  | 0.6406 | 0.3609 |
> |     | ✓      | ✓           | ✗         | ✓        | 24.33  | 0.6499 | 0.3578 |
> |     | ✗      | ✗           | ✓         | ✓        | 24.28  | **0.6500** | 0.3580 |
> |    | ✓      | ✓           | ✓         | ✓        | **24.35** | 0.6492 | **0.3571** |
> | MambaIR | ✗      | ✗           | ✗         | ✗        |24.35 |0.6587 |0.3244  |
> |     | ✗      | ✗           | ✗         | ✓        |24.73 |**0.6630** |0.3231  |
> |    | ✓      | ✓           | ✗         | ✗        |24.91  | 0.6616| 0.3198 |
> |    | ✗      | ✗           | ✓         | ✗        | 24.90 |0.6621 |0.3198  |
> |    | ✓      | ✓           | ✓         | ✗        |24.92  |0.6609 |0.3195  |
> |     | ✓      | ✓           | ✗         | ✓        | 24.93 |0.6620 |0.3200  |
> |     | ✗      | ✗           | ✓         | ✓        |24.92 |0.6629 |0.3201  |
> |    | ✓      | ✓           | ✓         | ✓        |**24.94** |0.6618 |**0.3194**  |
>
> **Table 7:** Ablation study of the $\tau$ weight used in the fine-tuning stage of pretrained SwinIR and MambaIR models.
> | Methods  | Metrics |$tau=0.1$|  $tau=1$|$tau=10$ |$tau=100$ |
> |-|-|-|-|-|-|
> | SwinIR | PSNR/SSIM/LPIPS |24.15/**0.6547**/0.3601 | 24.27/**0.6547**/0.3595 |24.30/0.6500/0.3596 |**24.35**/0.6492/**0.3571** |
> | MambaIR |  PSNR/SSIM/LPIPS |24.79/**0.6633**/0.3213 |24.84/**0.6633**/0.3211 |24.89/0.6632/0.3207 |**24.94** /0.6618 /**0.3196** |
>
> >  **Q6:** *Can you report multi-scale (×2/×3/×4) performance and discuss whether the LR-HF conditioning or noise schedule needs scale-specific adjustments?*
>
> **R:** Thanks. We report the performance of our LDP method applied to the ×2 SwinIR model pretrained for the RealSR task (available in the official SwinIR GitHub repository). For applying LDP at ×2 (or any other scale), no changes are required to the $LR_{hf}$ conditioning or to the noise schedule; the only component that needs to be modified is the Downsample Module. To apply LDP to any image-restoration task, we replaced the Downsample Module with a standard convolution, and the input HR image was replaced with the clean LR image $HR↓s$. We fine-tune the ×2 SwinIR model using this newly trained LDP. As shown in rebuttle Table 8, LDP also achieves competitive performance at the ×2 scale.
>
> **Table 8:** Performance improvements of SwinIR using our proposed LDP on synthetic multi-degradation benchmarks at $\times 2$ scale.
>
> | Datasets | Scale | Methods | NIQE | MANIQA | CLIPIQA | MUSIQ | QAlign|
> |- |- |- |- |- |- |- |- |
> | RealSR | x2 | SwinIR  |**4.789** |0.3380 |0.4794 |- |**4.005** |
> |   |x2 |  +LDP  |4.805 |**0.3484** |**0.5281** |-|3.999  |
> |DPED | x2 | SwinIR  |**4.700**| 0.2645| 0.3182|49.93 |3.139 |
> | | x2 | +LDP  |4.715| **0.2803**| **0.3626**|**52.29** |**3.226** |
> |RealSRSet | x2 | SwinIR  |6.070|0.4646 |0.6631 |65.65 |3.452 |
> | | x2 | +LDP  |**6.045**| **0.4747**| **0.6836**| **66.75**|**3.533** |
> | RealDeg | x2 | SwinIR |**3.709**|0.3265 |0.5379 |- |4.245 |
> | | x2 | +LDP   |3.742| **0.3281**| **0.5624**|- |**4.261** |
> | RealPhoto60 | x2 | SwinIR |**3.772**| **0.3791**|**0.5418** |56.01 |**3.061** |
> | | x2 | +LDP  |4.004|0.3572 |0.5241 |**57.21** |2.909 |

---

> ### Author Response · Authors · 2025-11-20
> **Official Comment by Authors (5/5)**
>
> >  **Q7:** *Please characterize training overhead more fully across backbones*
>
> **R:** Thanks. To provide more specific information, we extend the experiments in main text Table 14 (originally main text Table 12) by including a baseline using MambaIR, as shown in rebuttle Table 9. First, following the BSRGAN [1] setup, we train SwinIR and MambaIR from scratch. All four experimental groups use L1, LPIPS, and $L_{fre}$ losses (main text, Equation 13). In the LDP experimental group, we additionally employ $L_{sym}^{FT}$ (main text, Equation 15) as a loss function. We use DF2K as the training set. SwinIR is trained with a batch size of 8, and MambaIR with a batch size of 7, for a total of 100K iterations. The hyperparameter settings for LDP are as specified in Lines 311–312 of the main text: the key parameters are $s' = 2$, $L = 3$, $P = 16$, $Np = 32$, $\lambda_1 = \lambda_2 = 1$, and $C = 64$, resulting in 642k parameters. From the results, it is clear that using LDP as a loss provides an excellent trade-off between performance and efficiency.
>
> **Table 9:** Comparison of training cost and efficiency between the proposed LDP and baseline
>
> | Methods      | GPU memory (MiB) | Time per Iteration (s) | PSNR↑ | SSIM↑ | LPIPS↓ |
> |-|-|-|-|-|-|
> | SwinIR       | 15575            | 1.413                  | 23.64 | **0.6098**| 0.4541 |
> | SwinIR+LDP   | 22405            | 2.094                  | **23.96** | 0.6050| **0.4468** |
> | MambaIR       |  37021       |   1.575              | 23.94 | 0.6163 | 0.4358  |
> | MambaIR+LDP   |  44883        |  1.712                | **24.58** | **0.6343** | **0.4334**  |
>
> Ref:
>
> [1] Kai Zhang, et al. *Designing a Practical Degradation Model for Deep Blind Image Super-Resolution*. ICCV, 2021.
>
> >  **Q8:** *Could discuss scenarios where this reliance on LR high-frequency conditioning risks information leakage or trivial solutions, and propose safeguards or diagnostics?*
>
> **R:** Thanks. We employed several measures to ensure the absence of information leakage. First, at the input level, we do not use the LR image itself; instead, we use $LR_{hf}$, obtained by subtracting the LR image’s own downsample–upsample component from the original LR. Second, when receiving $LR_{hf}$, we introduce a prompt mechanism to further dilute its influence. In addition, within the denoiser design, the conditioning is applied through AdaIN, and we also inject patch-independent noise.
>
> We consider that trivial solutions in our LDP framework would correspond to cases where the model simply downsamples the HR image and adds $LR_{hf}$. To rule this out, we directly measure the metrics between this trivial reconstruction and our LDP outputs. The results are reported in rebuttle Table 10, and we also shows the similarity between LDP outputs and the LR inputs. Quantitatively, it is evident that LDP does not converge to trivial solutions.
>
> Therefore, the diagnostic we propose is to compute (1) the similarity between the LDP output and the HR-downsampled-plus-$LR_{hf}$ reconstruction, and (2) the similarity between the LDP output and the LR input. If the former becomes higher than the latter, it indicates that the learning process is problematic.
>
>
> **Table 10:** LDP Performance on Synthetic Multi-Degradation Datasets (×4 Scale)
>
> | Method | Scale | Datasets | Similarity to Trivial Reconstruction | LR Prediction Performance |
> |-|-|-|-|-|
> ||||PSNR/SSIM/LPIPS |PSNR/SSIM/LPIPS |
> | LDP | ×4 | Down   | 21.05 / 0.7507 / 0.2315 | **29.15 / 0.9283 / 0.0985** |
> | LDP | ×4 | Noise  | 18.86 / 0.7096 / 0.2657 | **26.71 / 0.8978 / 0.1248** |
> | LDP | ×4 | Blur   | 22.72 / 0.6344 / 0.4013 | **28.41 / 0.9159 / 0.1417** |
> | LDP | ×4 | JPEG   | 19.36 / 0.7697 / 0.2382 | **28.01 / 0.9243 / 0.0877** |
> | LDP | ×4 | Hybrid | 20.53 / 0.6383 / 0.3637 | **27.94 / 0.9173 / 0.1025** |

---

> ### Author Response · Authors · 2025-11-27
> **Looking forward to discussion**
>
> We sincerely appreciate the time and effort you have invested in reviewing our submission. We have carefully addressed your comments and made corresponding revisions to the manuscript. In addition, we have incorporated feedback from the other reviewers, which we hope will help resolve any further questions you may have.
>
> As the discussion period is ending soon, we wanted to follow up and would be grateful for any additional feedback or concerns you might wish to share, so that we can address them promptly.
>
> Thank you once again for your valuable time and dedication to the review process.

---

### Official Review · Reviewer_ceoH · 2025-11-04

**Soundness:** 1
**Presentation:** 2
**Contribution:** 1
**Rating:** 2
**Confidence:** 5

**Summary:**

The paper proposes LDP, a lightweight denoising autoencoder (DAE) designed as a plug-in module to improve the generalization of single-image super-resolution (SISR) models. It addresses the challenge of poor performance on unseen real-world degradations by modeling the degradation process and enforcing low-resolution (LR) cyclic consistency.

**Strengths:**

1. The proposed LDP introduces a lightweight, high frequency degradation concentrated degradation model that generates LR images from HR inputs.
2. The proposed LDP can be applied both during training and inference, enhancing generalization across diverse SISR models without architectural changes.

**Weaknesses:**

1. In the degradation prediction module, $LR_{hf}$ is obtained by subtracting the s’-fold downsampled-then-upsampled LR image from the original LR image. Although downsampled operation will corrupt a part of high frequency information, it also damaged the low frequency information. This approach to obtain high frequency component does not make sense.
2. When fine-tuning SR models with LDP, this paper claims to minimize the symmetric loss, which measuring the consistency of high frequency componets between LR input and a reconstructed LR from the SR output. When the reconstruction model (LDP) is freezed, however, the reconstruction consistency should decrease as the SR quality increases because better SR should remove more high frequency degradations from LR. Therefore, the symmetric loss should be maximized instead of minimized during fine-tuning and the proposed LDP can work like a degradation discriminator for the pretrained SR models.
3. In Table 1, there's only LDP's performance on synthetic degradation predictionon. Given no compared method existing, I cannot judge the effiency of the proposed method.
4. In Table 2, the performance enhancement of LDP is not significant. The improvement is especially trivial for the most updated SOTA method MambaIR and even MambaIR was published over one and a half years.
5. All visual comparisons are too small to distinguish the difference even after zoom in!

**Questions:**

1. What is the training purpose of the proposed LDP (Lightweight Denoising Plugin) framework? It seems like the authors want to predict a y' (or LR') with similar high frequency components of the input LR, but what is the necessity to train a network instead of using wavelet transformation or other frequency-based approaches to performe a high-pass filter on input LR? If LDP does not have LR as input and only use LR as supervision, this framework will be much more reasonable.
2. What is the purpose of using HR in the proposed LDP framework? Since $LR_{hf}$ provides the high-frequency information, it seems to exploit HR to gain more low-frequency or content information. However, this paper apply a blur kernel and the noise adding schedule from diffusion models in the LDP, producing a super degraded HR before entering the denoiser. Note the proposed denoiser adopt a lightweight structure so it cannot provides a similar denoising ability compared to the U-Net in diffusion models. Therefore, I doubt how much content information from HR can be provided for predicting the LR'.

---

> ### Author Response · Authors · 2025-11-20
> **Official Comment by Authors  (1/4)**
>
> We sincerely appreciate the time and effort you have devoted to reviewing our manuscript.
>
> From the review, it appears that LDP was interpreted as an LR high-frequency extractor or a deterministic degradation operator, and that the cyclic consistency loss was understood as enforcing the SR model to replicate LR degradation patterns. However, this appears to be a **misunderstanding**.  LDP narrows the SR solution space by introducing a constraint to enforce LR-domain consistency. It ensures that the SR output of any SR model can be degraded back to an LR image corresponding to the given LR input, following the widely adopted degradation-modeling paradigm (Section 2.2). Our design aligns with prior work such as DRN [1] and DualSR [2]. The key difference is that our LDP integrates degradation modeling~\cite{BSRDM} directly into the denoising autoencoder, reinterpreting denoising as a controllable degradation applied to HR images. Consequently, the Noise Addition Module (NAM) is an integral component of the LDP network and is applied consistently across all stages, including both LDP pretraining and inference. NAM injects noise into the features so that the noisy HR features are aligned with the LR features. Operations on these noisy HR features follow the classic degradation equation $LR = ((HR + n) \otimes K) \downarrow_s$, producing an LR image from the HR. Thus, LDP focuses on HR degradation rather than extracting high-frequency components from LR images.
>
> We have revised the manuscript by adding a **new Section 3.1 Motivation**, which explains the underlying principles behind our model design.
>
> Ref:
>
> [1] Yong Guo, et al.Closed-loop matters: Dual regression networks for single image super-resolution. CVPP, 5406–5415, 2020.
>
> [2] Mohammad Emad, el al. DualSR: Zero-shot dual learning for real-world super-resolution. In IEEE Winter Conference on Applications of Computer Vision, 1629–1638, 2021.
>
> With these clarifications, we provide detailed, itemized responses below.
>
>
> > **Q1:** *What is the training purpose of the proposed LDP framework? If LDP does not have LR as input and only use LR as supervision, this framework will be much more reasonable.*
>
> **R:** Thanks. LDP narrows the SR solution space by introducing a constraint to enforce LR-domain consistency. It ensures that the SR output of any SR model can be degraded back to an LR image corresponding to the given LR input. More specifically, LDP takes two inputs: the HR (or SR output) and a conditional $LR_{hf}$. It learns the degradation process to produce an LR output, and we perform regression to align this generated LR with the ground-truth LR. In the LDP training, we focus on supervising the high-frequency DWT subbands. However, our ablation studies in Appendix Table 9 (originally Appendix Table 7) show that supervising the low-frequency subbands or the entire LR image also yields similar performance.
>
>
> >  **Q2:** *What is the purpose of using HR in the proposed LDP framework? I doubt how much content information from HR can be provided for predicting the LR' after producing a super degraded HR before entering the denoiser.*
>
> **R:** Thanks. LDP degrades HR image into the LR image, ensuring that during inference, the SR result can be consistently mapped back to the input LR. Without an HR (or SR result) input, LDP has nothing to degrade.
>
> Our training process differs from that of diffusion models. In diffusion models, noise is added to the HR image in the forward process, and the model performs denoising in the reverse process to reconstruct the HR image; during inference, they start directly from pure Gaussian noise and only perform denoising. In contrast, LDP generates degraded LR images from HR images, and the degradation applied to the HR features follows classical degradation modeling. Consequently, LDP performs noise addition in both training and inference stages. Our denoising principle is inspired by diffusion-based methods [1] showing that noise injection can bring HR and LR features into alignment. Unlike [1], which adds noise to LR features to approximate HR features, we add noise to HR features to obtain LR features, making our training task easier. Operations on these noisy HR features follow the classic degradation equation $LR = ((HR + n) \otimes K) \downarrow_s$, producing an LR image from the HR images.
>
> Evidence from **main text Table 1 and Figure 3** further supports this interpretation: the HR input clearly provides the dominant content information for predicting the LR output, whereas  $LR_{hf}$ alone is insufficient to supply such content.
>
> Ref:
>
> [1] Zhixin Wang, et al. DR2: diffusion-based robust degradation remover for blind face restoration. CVPR, 1704–1713, 2023.

---

> ### Author Response · Authors · 2025-11-20
> **Official Comment by Authors  (2/4)**
>
> For the Weakness section:
>
> >  **Q3:** *In the degradation prediction module, to obtain high frequency component of the LR image by substracting its downsample then upsampled parts does not make sense.*
>
> **R:** Thanks. In general, downsampling acts as a low-pass filter. Therefore, the quantity obtained by subtracting the s'-fold downsampled-then-upsampled LR image from the original LR image captures most of the LR high-frequency residuals. We refer to this residual as *the high-frequency component of the LR image*, which we consider reasonable.
>
> >  **Q4:** *The reconstruction consistency should decrease as the SR quality increases.*
>
> **R:** Sorry for your misunderstanding. Our LDP is not designed to act as a degradation discriminator for pretrained SR models. It is not merely a downsampling operation applied to the SR image. LDP is specifically trained to reintroduce the degradation corresponding to the input LR image, functioning as a conditional degradation model. As the SR quality increases, the SR output becomes closer to the ground-truth HR image and better matches the LDP training. Consequently, the LR image produced by LDP from such an SR image will naturally be more similar to the input LR image. Therefore, reconstruction consistency should increase, not decrease, and the symmetric loss should be minimized.

---

> ### Author Response · Authors · 2025-11-20
> **Official Comment by Authors  (3/4)**
>
> >  **Q5:** *In Table 1, there's only LDP's performance on synthetic degradation predictionon. Given no compared method existing, I cannot judge the effiency of the proposed method.*
>
> **R:** Thanks. To demonstrate the superiority of our model under various degradation conditions, we provide a comparison between LDP and other degradation models on LR prediction. We compare against DRN [1] and DualSR [2]. DRN [1] adds a degradation branch that projects SR outputs back to the LR domain to enforce reconstruction consistency and improve stability. DualSR [2] introduces a dual-path framework in which a GAN-based downsampler and an upsampler are jointly trained with cycle consistency to model and reverse image-specific degradations.
>
> In this experiment, we use the same datasets as in main text Table 1. Five types of synthetic degradations are included: (1) downsampling, (2) noise, (3) blur, (4) JPEG compression, and (5) hybrid degradations following the BSRGAN-plus defaults. We first generate SR images using SwinIR, and then apply the degradation models provided by DRN, DualSR and LDP to obtain the predicted LR images from the SR outputs. These predictions are then compared with the LR inputs to the SR model, and the results are reported in rebuttle Table 1. In addition, rebuttle Table 2 reports the similarity between the LR images produced by the degradation models and the downsampled SR images produced by SwinIR. A higher similarity indicates that the degradation model fails to apply the specific degradations implied by the input LR and instead behaves like a simple downsampler.
>
> As shown in the tables, our LDP performs consistently well across different degradation types. The similarity between the LDP-generated LR and the downsampled SR is significantly lower than that between the LDP-generated LR and the input LR, demonstrating that LDP does not collapse into trivial downsampling. In contrast, DRN behaves almost identically to bicubic downsampling. Its inputs include only HR/SR images without any conditional information, so it cannot correctly map an SR image to multiple possible LR variants. Its similarity to the downsampled SR image is much higher than its similarity to the LR input. DualSR appears to struggle with handling diverse degradation types.
>
> We have revised Section 4.2 of the manuscript. The changes are temporarily highlighted in blue for ease of review.
>
> **Table 1:** Performance of multiple degradation models in LR prediction on synthetic multi-degradation datasets.
> |Method|Down|Noise|Blur|JPEG|Hybrid|
> |-|-|-|-|-|-|
> ||PSNR/SSIM/LPIPS|PSNR/SSIM/LPIPS|PSNR/SSIM/LPIPS|PSNR/SSIM/LPIPS|PSNR/SSIM/LPIPS|
> |DRN|**32.05**/**0.9539**/**0.0794**|**27.25**/0.7812/0.2474|26.38/0.8273/0.3207|**29.65**/**0.9270**/**0.0826**|27.03/0.8098/0.3360|
> |DualSR|19.58/0.4814/0.1408|18.77/0.4712/0.1399|19.36/0.4911/0.1844|18.57/0.4612/0.1492|19.36/0.4883/0.2130|
> |LDP|29.15/0.9283/0.0985|26.71/**0.8978**/**0.1248**|**28.41**/**0.9159**/**0.1417**|28.01/0.9243/0.0877|**27.94**/**0.9173**/**0.1025**|
>
>
> **Table2:** Similarity between the LR images generated by multiple degradation models and the downsampled SR images.
> |Method|Down|Noise|Blur|JPEG|Hybrid|
> |-|-|-|-|-|-|
> ||PSNR/SSIM/LPIPS|PSNR/SSIM/LPIPS|PSNR/SSIM/LPIPS|PSNR/SSIM/LPIPS|PSNR/SSIM/LPIPS|
> |DRN|**34.02/0.9638/0.0365**|**31.57/0.9590/0.0436**|**34.99/0.9692/0.0306**|**31.35/0.9587/0.0467**|**35.10/0.9679/0.0296**|
> |DualSR|22.58/0.6689/0.1264|20.79/0.6502/0.1040|22.57/0.7044/0.1262|20.46/0.6356/0.1279|22.85/0.7164/0.1175|
> |LDP|28.41/0.8895/0.1551|25.93/0.7508/0.3043|25.04/0.7596/0.3278|27.42/0.8886/0.1293|26.28/0.7597/0.3586|
>
> Ref:
>
> [1] Yong Guo, et al.Closed-loop matters: Dual regression networks for single image super-resolution. CVPP, 5406–5415, 2020.
>
> [2] Mohammad Emad, el al. DualSR: Zero-shot dual learning for real-world super-resolution. In IEEE Winter Conference on Applications of Computer Vision, 1629–1638, 2021.

---

> ### Author Response · Authors · 2025-11-20
> **Official Comment by Authors  (4/4)**
>
> >  **Q6:** *In Table 2, the performance enhancement of LDP is not significant.*
>
> **R:** Thanks. The performance gains introduced by LDP to existing SR models, as shown in main text Table 3 (originally main text Table 2), are clearly evident. From the table, we can see that for MambaIR under the bicubic degradation scenario, LDP improves PSNR by 0.05 and SSIM by 0.001. As observed with MambaIRv2 [1], achieving comparable gains under the bicubic degradation scenario typically requires a new model design, larger parameter capacity, and over 500k training iterations. In contrast, our approach only adds LDP as an additional loss to the original model and requires merely 1,000 fine-tuning iterations. Moreover, the improvements are even more significant across other models and various degradation scenarios; for instance, the PSNR gain for MambaIR under the hybrid degradation scenario reaches as high as 0.36. To make these performance gains more explicit, we have revised main text Table 3 (originally main text Table 2) in the manuscript to highlight the improvements for each metric.
>
> Ref:
>
> [1] Hang Guo, et al. MambaIRv2: Attentive State Space Restoration. CVPR, 28124-28133, 2025.
>
> >  **Q7:** *All visual comparisons are too small to distinguish the difference even after zoom in!*
>
> **R:** Thanks. **Figures 4 (main text)** clearly demonstrate our ability to restore parrot feather textures and recover fine fabric details, particularly highlighting the visual improvements over StableSR. **In Figure 5 (main text)**, we suppress ringing artifacts produced by the baseline model, enhancing its visual quality. **Figure 6 (main text)** shows how we mitigate incorrect texture restorations of mountains and buildings by the baseline. **Figures 8 and 9 in the Appendix** provide additional visual comparisons, further emphasizing the effectiveness of our method in suppressing artifacts generated by the baseline.

---

> ### Author Response · Authors · 2025-11-27
> **Looking forward to discussion**
>
> We sincerely appreciate the time and effort you have invested in reviewing our submission.  We noticed there may have been some misunderstanding regarding our LDP method and have provided clarifications in our rebuttal. Please let us know if our response has adequately addressed your concerns.  In addition, we have incorporated feedback from the other reviewers, which we hope will help resolve any further questions you may have.
>
> As the discussion period is ending soon, we wanted to follow up and would be grateful for any additional feedback or concerns you might wish to share, so that we can address them promptly.
>
> Thank you once again for your valuable time and dedication to the review process.

---

### Author Response · Authors · 2025-11-20
**Revision Note to Reviewers and Area Chair**

We thank the reviewers for their constructive feedback.
Based on the comments, we have revised the manuscript accordingly.
All modifications are temporarily highlighted in blue in the updated submission.

**Major changes include:**

- **Figure 1 revised** to more clearly illustrate that the purpose of LDP is to enforce LR consistency.

- **A new Section 3.1 (Motivation) added** to explain the design principles behind LDP and clarify its role as a conditional HR→LR degradation model.

- **Figure 2 updated**, where the inputs in Fig. 2(c) and 2(d) have been replaced with feature maps instead of images to avoid confusion.

- **Section 4.2 revised** to include additional degradation-modeling baselines (RDN and DualSR).
Corresponding updates were made to Table 1, and we added Table 2 and Figure 3 for clearer comparison.

- **Tables 3–5 updated** (previously Tables 2–4) to explicitly report the performance gains brought by LDP over each SR backbone.

- **Ablation on the loss weight $τ$ added** in Section 5, with a new Table 7 summarizing the results.

We hope these revisions improve the clarity and completeness of the paper, and we thank the reviewers again for their time and valuable suggestions.

---

> ### Author Response · Authors · 2025-12-01
> **Final Changes**
>
> We have submitted the final revision. The portions previously highlighted in blue have been reverted to black, and no new content has been added beyond these changes.

---

### Author Response · Authors · 2025-12-01
**Summary for AC**

We sincerely appreciate the time and effort the AC has devoted to reviewing our manuscript. To help the AC **quickly grasp** the core of our work and the key points of our rebuttal, we provide the following **brief summary**.

In this paper, we propose LDP, a plug-in that models the super-resolution degradation process within a denoising autoencoder (DAE) framework. LDP takes an HR image as input (either ground truth or the SR model’s output) and predicts its corresponding LR image. With conditional information, LDP can produce different LR variants from the same HR image. This allows our method to better handle diverse real-world degradations. LDP can be applied to SR models in two modes: as a training loss to improve reconstruction quality, or as an inference post-processing step to correct artifacts.

Before the discussion phase, we received two scores of 6 and one score of 2.

> Reviewer ceoH (Score: 2 / Confidence: 5) **misinterpreted** our LDP, misunderstanding it as either a high-frequency filter applied directly to LR inputs or a module that does not require the SR (HR) results as input. What we need to clarify is that LDP serves as a degradation model that further degrades the SR output back to an LR image. Therefore, our model must receive the SR result or the HR image as input; otherwise, there would be no target on which the degradation operation could be performed. Due to this misunderstanding of LDP’s functionality, reviewer ceoH also suggested that the symmetric loss (the distance between the predicted LR and the ground-truth LR) should be increased rather than minimized, which clearly contradicts our goal. We clarified this misunderstanding in the rebuttal.

> Reviewer hH62 (Score: 6 / Confidence: 4) considered **our LDP to have strong significance for the SR community** and requested more types of SR baselines for ablation beyond the SwinIR results we initially provided. We conducted and submitted all the additional experiments requested during the rebuttal.

> Reviewer C4zd (Score: 6 / Confidence: 4) acknowledged that our LDP is lightweight, plug-and-play, and **generalizable** to most SR models. Reviewer C4zd asked us to further demonstrate the generalization of LDP under more extreme environments and to other image restoration tasks, as well as the performance under JPEG compression. We responded to all these concerns in detail during the rebuttal.

Following the reviewers’ suggestions and questions, we **revised** the manuscript accordingly. The major updates include (1) the addition of Section 3.1 Motivation, (2) new comparisons on LR prediction performance between LDP and other methods, and (3) a more direct presentation of numerical improvements in the tables. We hope these revisions further improve the readability of our manuscript.

We would like to express our sincere gratitude to the reviewers and ACs once again for their thorough evaluation and insightful suggestions, which have helped us improve the quality of this work.

---

### Meta-Review · Area_Chair_sSkB · 2026-01-08

**Summary:**

This paper proposes LDP, a lightweight denoising autoencoder plug-in designed to improve the generalization of SR models via LR-prediction-based cyclic regularization. The core idea of using a degradation model as a regularizer is clear and the experimental scope is extensive. This is primarily due to the significant concerns raised by Reviewer ceoH regarding the fundamental soundness and novelty of the method, which were not fully resolved. The rebuttal clarified a key misunderstanding about LDP's role as a degradation model (not a high-pass filter), and authors provided valuable additional comparisons against methods like DRN and DualSR, as requested by other reviewers. Nonetheless, doubts remain about whether the proposed approach offers a substantial theoretical or practical advance over existing degradation modeling and cycle-consistency concepts in SR. The improvements, while consistent, are often marginal, and the paper's central premise did not convince all reviewers of its necessity and distinct contribution.

**Reviewer Concerns:**

The rebuttal addressed several specific technical concerns: it clarified LDP's function as an HR→LR degradation model (ceoH), added direct comparisons to prior degradation-modeling works like DRN and DualSR (hH62, C4zd), extended evaluations to more real-world datasets (hH62), and provided ablations on loss weights and severe blur conditions (hH62, C4zd). However, major concerns remain outstanding: 1) The core novelty and necessity of LDP within the landscape of existing cycle-consistency and degradation-modeling methods is still questioned (ceoH). 2) The theoretical justification for why this specific DAE formulation generalizes well to unseen degradations, particularly those not explicitly modeled in its training (like JPEG, per C4zd's follow-up), lacks depth and relies heavily on empirical results. 3) Questions about the significance of the gains and the method's practical impact, given the sometimes modest improvements, are not fully alleviated.

**Reviewer Scores:**

With full discussion, ceoH (score: 2) might have raised their score to a 4 (borderline). The rebuttal clarified the critical misunderstanding about LDP's input/output, potentially moving it out of the "poor soundness" category, but their fundamental concerns about novelty and contribution significance would likely persist, preventing a score above the acceptance threshold. hH62 (score: 6) and C4zd (score: 6), who were already marginally positive, would likely have increased their scores to a 7 (clear accept). Their requests for more baselines (DRN, DualSR) and evaluations on additional datasets/extreme conditions were comprehensively met with new experiments in the rebuttal, strengthening their confidence in the paper's empirical contribution and plug-and-play utility.

---

### Decision · Program_Chairs · 2026-01-26

Reject